# Effects of residential mobility and migration on standards of living in Dar es Salaam, Tanzania: A life-course approach

**Amit Patel**[1]*, **George Joseph**[2], **Namesh Killemsetty**[1], **Sokha Eng**[1]

**1** Department of Public Policy and Public Affairs, McCormack Graduate School of Policy and Global Studies, University of Massachusetts, Boston, MA, United States of America, **2** Water for South and East Asia Unit (Water for SEA), Water Global Practice, World Bank, Washington, DC, United States of America

☉ These authors contributed equally to this work.
* Amit.Patel@umb.edu

**Data Availability Statement:** Dataset used for this study is publicly available from The World Bank Group's website. Following details will provide other researchers the same access to data as the

## Abstract

Rural-to-urban migration and intra-urban residential mobility often lead to improved living conditions. However, it is not clear if this is true for all, especially in cities in developing countries, where inequalities persist and upward mobility remains elusive for marginalized populations. We investigate the effects of intra-urban residential mobility and rural-to-urban migration on standards of living in Dar es Salaam, Tanzania. We examine the entire residential history of individuals and assess temporal changes in living conditions in their respective housing careers. We use four different aspects of housing as indicators of living conditions from a survey conducted by the World Bank Group that captured the residential history of 2,397 individuals in Dar es Salaam from a spatially representative sample of households. We find that improvements in housing conditions are uneven, a considerable proportion of individuals remained perpetually deprived of adequate housing despite migrating from rural areas to Dar es Salaam and changing multiple homes within the city during their tenure. Our findings indicate that housing disparities persist over time for many. Socioeconomic groups such as migrants tend to experience significant improvements after moving to the city but show limited upward mobility afterwards, an aspect that is rarely addressed in policy discourse on equitable access to adequate housing.

## Introduction

For a long time, it was assumed that residential mobility was a pathway for people to move to better homes and neighborhoods, leading to better socioeconomic and health outcomes. However, life-course studies have recognized that not all residential moves are a result of choice, and they may not always result in better outcomes for everyone. Most of this research has been conducted in the developed world, particularly in the United States (US), United Kingdom (UK), and Europe, where large-scale longitudinal datasets are available to study why people move and how such moves affect a wide variety of outcomes [1–8]. However, the effect of residential mobility on housing conditions is rarely studied in the context of the developing world

authors. Study description, documentation, data description and microdata are available publicly and freely from the URL provided below. Specific survey ID and title are provided for reference. Dataset is adequately cited in the manuscript as well. URL: https://microdata.worldbank.org/index.php/catalog/3399 Title: Measuring Living Standards within Cities, Dar es Salaam 2014-2015 SURVEY ID NUMBER: TZA_2014_DAR-LSMS_v01_M.

**Funding:** AP received funding from the World Bank Group via Research Agreement No. 7186069 (Funder website: https://www.worldbank.org). GJ received funding via salary from the World Bank Group. The funder had no role in design, analysis, or decision to prepare and publish this manuscript. The specific roles of these authors are articulated in the 'author contributions' section.

**Competing interests:** The authors have declare that no competing interests exist.

in general, and in the context of slums in particular. This is partly due to the limited availability of comprehensive and longitudinal datasets for cities in the developing world and a nearly complete absence of such data for slums. This paper makes a significant contribution to the scant literature by highlighting the impact of rural-to-urban and intra-urban mobility on the living conditions of both slum and non-slum households in Dar es Salaam, Tanzania. There is no dearth of cross-sectional studies on slums that focus on health [9–14] and housing conditions [15, 16], but none of the literature on slums has focused on the residential trajectories with some notable exceptions [17–19]. The study [17] focused on slum-dwellers' revealed preferences for location quality over housing quality by examining residential history. The works by [18, 19] are among the first to explore the aspirations of migrants in Dar es Salaam and how they are related to their residential mobility using case studies and focus group discussions with residents of select communities. We add to this growing but nascent body of literature and complement it with the first application of a life-course approach to investigate the impact of residential mobility on housing conditions in the context of the developing world.

We strongly believe that such empirical work is useful in generalizing housing theories from the developed world as well as drawing implications for international development policies that focus on housing and such basic services as water and sanitation. For example, slum policies commonly focus on deprived neighborhoods in the global South, including Dar es Salaam [20], but rarely examine the temporal dimension of deprivation. In rare instances when the temporal dimension has been addressed, such as using a fixed cut-off date as an eligibility criterion for slum redevelopment projects in Mumbai, they have made slum policies exclusionary [21]. Our plea in this paper is to use temporal aspects of housing deprivations to make slum policies more inclusionary and just. A better understanding of housing moves could also provide insights into how such moves may create unstable housing environments for residents, especially because housing stability continues to remain a tacit goal behind all home ownership-oriented policies [22].

The literature on housing dynamics in the European and US contexts has shown that residential mobility can be beneficial for both residents and the housing market. While residents generally get better housing units and neighborhood amenities as a direct consequence of such moves, residential mobility can also lead to adverse outcomes [1, 23, 24], depending on what has motivated or triggered the housing move. Similarly, it is quite likely that housing moves for the urban poor in the developing world do not always result from choices driven by an intent to improve their living conditions. Often, they might be forced by events such as push factors in rural origins (e.g., drought) or financial shocks (e.g., loss of employment) that may force them to accept inferior housing conditions within a city. Worse, slum-dwellers can be victims of larger housing market forces, development project-induced relocation, or housing policies that lead to evictions in slums to enforce *de jure* property rights that slum-dwellers often lack. As [19] has shown, aspirations of homeownership often lead to accepting lower living standards. In this study, we examine the life-course of households to explore the effects of rural-to-urban migration and residential mobility on residents' housing conditions in Dar es Salaam, Tanzania, using sequence analysis techniques applied to a unique dataset that captures the entire residential history of 2,397 individuals from the sampled households.

The main objective of the paper is to systematically examine housing moves and how they relate to change in housing conditions for individuals. There are three distinct advantages of examining the entire residential histories of individuals. First, it overcomes the limitations of cross-sectional studies and provides insights about the most common pathways that residents follow as they experience improvement or decline in housing conditions [25]. Second, such an approach does not assume independence between events, and it explicitly recognizes the path dependence of housing moves. Third, the life-course approach connects individuals with the

structural realities of the housing market that shapes their opportunity structure and influences what kind of housing conditions they are likely to experience over their lifetime as a function of such structural factors as well as their individual characteristics.

## Literature review

We briefly summarize classical theories of migration and residential mobility that focus on the motivations behind such moves. Subsequently, we review the life-course approach and its applications to study housing trajectories, which are commonly referred to as housing careers in the literature. It is clear from this review that predominant migration theories focus on labor migration and wage differentials associated with labor markets, whereas life-course approaches study intra-urban residential mobility. However, both sets of theories rarely focus on the impacts of such movements, including migration or intra-urban mobility, on such housing outcomes as access to water and sanitation, durable structures, or homeownership, which are a central focus of our research presented in this paper. Finally, this review makes it clear that there is no dearth of such studies in the context of the US, UK, and Europe, but these theories are rarely tested in the context of developing countries.

## Why do people move?

It is essential to distinguish between migration and residential mobility. While both involve changing a home, migration is a move from one labor market to another, constituting a move from one region to a completely different region over a long distance, whereas residential mobility is defined as intra-urban moves over relatively short distances within the same region [26]. Traditionally, intra-urban residential mobility is associated with housing-related reasons, while migration is motivated by such non-housing reasons as a job change in pursuit of better economic opportunities [27, 28].

Several migration theories explain why people migrate [29–44] and present several explanatory factors, such as rural-to-urban wage differences, geographic distances between rural origins and urban destinations, culture of migration among certain communities, push factors in rural origins, and pull factors in urban destinations, to name a few. Migration studies often focus on the consequences of destinations at the aggregate level, including migration's effects on labor market equilibrium [45, 46] and demographic convergence [47]. Inter-regional migration is believed to lead to improvements in housing and neighborhood conditions [1], financial gains [48], and subjective well-being [49]. However, it has been observed that these improvements are uneven, and some migrants suffer from such moves, including such non-economic losses as weakened friendship and family ties [23, 29, 50]. Dominant gender roles and identities also play a crucial role; the benefits of migration are often limited only to the men in a family, while married women support their husbands' careers by following them as tied migrants independent of their occupational status [51]. In the global South in general, and in the African context in particular, rural-to-urban migration is often a family survival strategy, with diverse impacts on migrants in the city and the family members left behind in their rural origins [52].

There may be multiple reasons for families to move within a city. Positive reasons include getting better job opportunities, moving closer to work to reduce commuting times, building families, buying new homes, moving closer to extended families, and seeking new neighborhoods with better schools. Conversely, negative reasons may include divorce, death, domestic violence, eviction or foreclosure, forced relocation, or diminished financial resources from loss of employment [53].

The consequences of intra-urban mobility have been studied in multiple contexts in the developed world. Researchers have reported on how such moves can lead to improvements in labor market outcomes, social welfare benefits, or provisions of childcare [24], as well as leading to negative consequences [5, 53]. Studies also show that intra-urban residential mobility is generally motivated by housing-related factors, whereas inter-regional migration is influenced by employment opportunities or family considerations [54–56]. Within a city, low-income families have higher rates of residential mobility than middle- and upper-income families, and their moves are less likely to be for positive reasons [53, 57]. Studies have highlighted the need to understand consequences of migration and intra-urban mobility on housing conditions [1]. Our study is an attempt to capture such transitions and understand their effects on the quality of housing conditions.

## Life-course approach to studying housing careers

The life-course approach has emerged as a defining way to conceptualize and organize the complex changes that individuals experience as they age over time. It is commonsensical that individuals do not get married, have children, or change jobs in a lock-step manner with age; instead, they greatly vary in behavior related to having children or changing jobs. Some have more or fewer children sooner or later, and some change their jobs more frequently while others remain in a single job for decades. The life-course approach recognizes such diverse behaviors and de-standardization and captures them in a rich way that cross-sectional studies often tend to ignore [58, 59]. A life-course has also been defined as a sequence of roles and events that sometimes become turning points [60]. Such a definition establishes life-course as a concept that emphasizes the trajectory of individuals as they progress from initial entries into labor or housing markets to their subsequent moves and adjustments as they move up (or down) in labor or housing markets. While time is related to such trajectories, all individuals make such transitions at varying stages in their lives, displaying a wide variety of sequencing and timing in transitions in job, family, or housing careers. Separating these events and sequencing them to understand the life course of an individual provides an excellent framework for studying changes in living conditions. Furthermore, individuals experience different housing outcomes purely as a result of cohort and period effects. Residents who have experienced a city in a period when housing conditions were poor are less likely to have good housing conditions compared to those who were in later cohorts [60].

As discussed earlier, there are two general types of housing dynamics studies: i) studies focusing on decisions to move, moving processes, and the factors affecting such decisions, and ii) studies focusing on outcomes of such moves on various aspects of socioeconomic wellbeing, including implications for health, education, and wealth creation. The first type of studies tends to focus on issues of timing, such as how events in other domains of life trigger specific residential moves. For example, the birth of a first child that transitions a couple into a family is closely tied to a residential move that immediately precedes or follows the birth event [61]. Similarly, the addition of a new earner in a household (such as after a member moves in after getting married to a working person) that results in a substantial increase in income has been associated with a residential move [2]. Studies have also shown direct linkages between changes in family composition and jobs and improvement in dwelling quality [3], also demonstrating how housing moves can be associated with housing market booms and busts and housing policy changes at the macro-level [62].

The second type of housing dynamics studies has focused on implications of residential moves on access to a variety of opportunities, such as income mobility and access to jobs, better healthcare, and education services, to name a few. For example, studies have shown the

effects of residential moves on health outcomes for individuals [63, 64]. It is this tradition of examining outcomes as a direct consequence of residential moves that motivates our study. We examine the effects of both migration and intra-urban residential mobility to report changes in housing conditions related to basic services, specifically focusing on access to water and sanitation, durable structures, and tenure status.

## Research objectives

The extant studies on housing for the urban poor focus on cross-sectional and horizontal disparities between different sections of society. The longitudinal view presented in this paper could enhance our understanding of the temporal and dynamic nature of such inequalities. We believe that durable inequality exists between various classes of a society that perpetually deprives some classes over others [65]. In our opinion, there are at least three distinct benefits of having such a longitudinal and sequential view. First, most housing studies look only at a single move (such as a rural-to-urban migration and comparison between housing in the origin and the destination), thus limiting our understanding of periods of deprivation over time. Because people typically move several times in a life-course, a critical dynamic is missing from our current understanding of life in slums and associated changes in housing conditions. A longer view captures the uneven pattern of sequences of the housing states as people progress through their life-course in a city. Second, because the emphasis in the literature is on single moves, the quality of housing that is often achieved at the top (or end) of a housing career of an individual is rarely understood. The life-course perspective could at least tell us what paths people have followed to achieve the highest housing quality at the end or at any point in life, if they have achieved it at all. Third, housing careers are expected to vary across time and space. Looking at housing careers in various parts of a city could highlight differences in progression or stagnation of the housing careers as well as housing quality achieved in different parts within a city (e.g., city center versus periphery).

In light of this discussion, we posit that living conditions vary across individuals and that the experience of migrants is systematically different from that of non-migrants or native residents. Similarly, we expect that people living in shanties follow different housing careers than those living in non-shanties. Consequently, we investigate the following research questions:

1. What are the patterns of housing conditions over the life-course of individuals in Dar es Salaam?

2. How do housing careers differ between migrants and natives, and between slum and non-slum residents in Dar es Salaam?

## Data and methods

### Case study city: Dar es Salaam, Tanzania

Dar es Salaam, which is Arabic for 'Harbor of Peace,' was originally built around a large natural harbor in the 1800s [66, 67]. It has flourished as an administrative and commercial center under the German East Africa Company, British rule in the early 1900s, and its post-independence growth since 1961. By the 1950s, the British colonial administration in Dar es Salaam witnessed diverse pressures due to massive rural-to-urban migration that overwhelmed the colonial ideal of an orderly town [68]. The city has subsequently observed an enormous expansion over time [69]. According to the last available 2012 national census, the city had a population of over 4.3 million people with 3,133 people per square kilometer, making it the most

densely populated region in the country [70]. As the urban growth in the city accelerated, the rate of unplanned housing also accelerated [71, 72], with about 60% of the city's total urban population living in unplanned settlements [68]. A public housing program designed to meet housing demand was swamped by rapid urban growth [73]. Studies have highlighted how urbanization has been a key factor underpinning and catalyzing changes in land use, land transactions, increased rural-to-urban immigration, and the overall transformation of land use in the peri-urban areas in Dar es Salaam [74].

Dar es Salaam is home to 40% of Tanzania's urban population, with a large share living in low-income and slum settlements [75]. The growth of the city has been documented extensively [19, 76–78]. The city is primarily dominated by low-rise, low-density development, with the exception of the central business district [19]. The surface area of the city has dramatically increased, with the maximum distance from the center to the edge increasing from 6–10 km in 1969 [76] to 50 km in the early 2000s [79].

The population in Dar es Salaam has been growing exponentially in the peripheral areas, with the rural areas continuously being developed and incorporated into the urban landscape in the form of informal settlements or slums with poor infrastructure services [19, 76, 80]. The search for affordable housing is the primary motivation for residents to move towards the periphery for self-built, owner-occupied housing [19]. The majority of the peripheral settlements are un-serviced by networked infrastructure [81, 82]. With housing development taking place before the provision of services, access to central water and sanitation networks is limited, leading to families relying on on-site systems. Access to infrastructure services is mostly created, organized, and financed by residents with the assistance of various informal self-help solutions [18].

Informal settlements in Dar es Salaam are usually described as unplanned neighborhoods where a mix of middle- and low-income families live [83]. While the formal city still follows colonial planning regulations and building standards, the informal city has developed in between the main roads and at the periphery [83]. An estimated 80% of Dar es salaam's territory is informal [84], which is more accessible and affordable but offers less de facto tenure security [19]. However, it has been observed that households with houses built from permanent and modern building materials living in these informal settlements often perceive security of tenure [76]. Ownership in these settlements is mostly legitimized through informal sales agreements and social recognition from local leaders and landowners [19].

The private sector is estimated to own 95% of the total housing stock in urban areas in Tanzania [85]. The houses are built by private individuals who mobilize financial resources from varying sources. The houses supplied by this sector are mostly detached, one-family, self-contained or multifamily units with shared facilities like toilets, bathrooms, and cooking facilities. Most houses in the urban areas are occupied by both owner households and tenants who rent one or more rooms.

The provision of infrastructure services has not kept pace with the demographic and spatial growth of the city [86]. Water and sanitation services in African urban areas especially in Dar es Salaam are overstretched, with the primary networks inaccessible to everyone in the city [87]. The formal municipality usually delivers services in these non-planned settlements, but services are inferior and unreliable. The poor continuity of service ensures that households rely on informal service providers, which include mobile vendors, water tankers, private boreholes, and neighborhood resellers, among others. Relying on informal sources not only raises affordability concerns for the residents but also increases health risks because of the poor quality of water [88]. The responsibility for water supply and sanitation is divided between an asset-holding company that is responsible for capital investments (the Dar es Salaam Water and Sewerage Authority—DAWASA) and an operating company that runs the water and

sewer system on a day-to-day basis and bills customers (the Dar es Salaam Water and Sewerage Corporation—DAWASCO) [81].

Multiple studies have highlighted the problems faced by households in Dar es Salaam because of their limited access to such basic infrastructure services as water supply and sanitation [89–91]. Studies [87, 90] trace the history of water and sanitation provision in Dar es Salaam, highlighting the issues existing from the colonial period to the post-colonial period, when the inadequate systems had limited expansion or improvement in services despite significant population growth.

Access to sanitation services for the majority of the urban poor households in Dar is expensive, difficult to obtain, inappropriate, and unsafe [91]. The location of a neighborhood in the city plays a vital role in gaining access to water and sanitation. Only 3.7% of the city's households had access to the city's piped sewerage system in 2010, with an overwhelming 90% of the population relying on on-site sanitation systems or other kinds of domestic disposal [87, 92–94]. Responsibility for the operation of sanitation facilities (such as cleaning of public latrines and solid waste collection) is divided among multiple organizations at different levels, which face limited capacity to implement at a larger scale [88].

The governance of water supply and sanitation networks in Dar es Salaam has undergone several phases of restructuring, from the development networks in urban centers during the colonial period in the 1930s to their privatization in the early 2000s to improve operation and maintenance efficiencies [95]. The evolution of water policies has reflected a transition from the centralized and free provision by the state to a more centralized demand-responsive approach emphasizing cost recovery [88]. The urban poor of the city primarily rely on informal sources of water, such as tanker trucks or carts with small tanks [89]. Approximately 58% of the city's inhabitants are served by the existing centralized water system [87], with the remaining relying on informal, small-scale operators [82, 96, 97]. Therefore, urban populations end up adopting multiple formal and informal strategies to secure access for their daily requirements [87, 90].

## Data

This paper uses the survey data collected by the World Bank's Spatial Development of African Cities and Global Urban Data Program, which aimed to build knowledge about urbanization and support evidence-driven policymaking. In this section, we summarize the relevant details of the survey. The complete details and data are publicly available through the Microdata Library of the World Bank [98]. The survey instrument was designed to measure living standards within cities as a response to the growing need to understand urban living standards in Africa and across the world. The sampling strategy incorporated geo-referencing to collect spatially representative data within each city to draw conclusions about variations in living standards over space. The surveys were designed to provide new insights into housing and basic services and were conducted in Dar es Salaam between 2014 and 2015. The city was divided into three geographic stratums: i) the core city area, ii) the consolidated city area, and iii) the urban periphery. The core city area was further divided into shanty and non-shanty areas using very high-resolution satellite images. In the end, four different strata were identified. The detailed methodology that defines these stratums, including identification of shanty and non-shanty areas from satellite images, is available in Annex 1 of the accompanying documentation of the publicly released dataset [98]. While slums, informal settlements, and shanties, all refer to areas that are associated with poor housing conditions, definitional differences are not trivial [99–101]. For the purpose of this paper, we are using these terms interchangeably but refer to shanties as defined in the data documentation. For each of these stratums,

Enumeration Areas (EA) were selected as Primary Sampling Units (PSU), using the 2012 Census data as a sampling frame with the probability proportional to size (PPS) method. In the second stage, 12 households were randomly selected from each EA, resulting in a total of 2,400 sampled households from 200 EAs, of which 2,397 households responded to the survey.

Among multiple aspects of standards of living, the study collected entire residential histories of the head of each sampled household, starting from a respondent's home at birth for Dar es Salaam residents or from the last home in a rural origin in case of migrants (for the purpose of this study, migrants are defined as all individuals currently residing in Dar es Salaam who were born outside Dar). There were 7,009 homes occupied by 2,397 sampled households in total. Such a residential history is rarely collected in the context of developing countries, and seldom in the case of slum-dwellers. This is one of the first surveys that collected such a detailed residential history, including access to such basic services as water and sanitation, building materials used for dwelling, type of tenure, number of years spent at each residence, and reasons for moving out. This residential history allows us to apply the life-course approach to the data. Although the data was not collected by continuously monitoring residents for their entire life-course at set intervals, we believe that this recall data of discrete events (i.e., housing moves) accurately captures the residential history of individuals. The number of moves for a resident is usually no more than ten homes, and most respondents remembered such basic information as availability of water, sanitation, building materials, tenure type, duration of stay, and reasons for moving out for homes that they have personally occupied. However, out of 7,009 reported homes, there were gaps in information about access to water for 441 homes, in access to sanitation for 439 homes, in type of ownership for 435 homes, and in type of structure for 817 homes, resulting in a smaller sample size for respective analyses. It is important to note that despite these limitations, this history is considered a rather rich dataset in the developing-countries context.

## Methodology

Changing homes is an involved process that is not confined to a physical move from one geographical area to another. These moves affect and are affected by life-courses and further impact movers' housing conditions. In order to capture these changes in life-course systematically, sequence analysis provides an adequate analytical framework [25]. Sequence analysis techniques were originally developed to study structures and patterns in genes and genomes by biostatisticians but have been more recently used in social and cultural studies [102].

Sequence analysis is a useful tool to study processes or series of events that usually happen in a particular order. A sequence is an ordered list of items, which could be events, numbers, characters, or anything else where sequences matter. Beyond the simple question of the existence of patterns, sequence analysis helps answer other questions, such as what influences those patterns have on other variables and what other variables dictate the patterns that the cases follow [103]. Sequence analysis takes sequences of data as inputs rather than individuals as data points and uses similarities between sequences to understand regularities and patterns among them [104].

In terms of process, the methodology for sequence analysis consists of five steps: i) *description*, with tabulation of sequences and calculation of indicators for characteristics of each sequence; ii) *visualization*, with sequence index plots and parallel coordinates plots; iii) *comparison* using distance measures obtained via optimal matching; iv) *grouping* of similar sequences based on the results of comparisons using techniques like cluster analysis; and v) *application* by using grouped sequences with dependent or independent variables in standard regression models [105]. We follow the first two steps of this taxonomy since our primary

objective is to explore the patterns in sequences of housing conditions in Dar es Salaam. We modify the third step and compare patterns in sequences in our prespecified groups of interest, namely differences between migrants and non-migrants and differences between those who live in shanties and those who do not. We do not undertake fourth and fifth steps, as they are not directly linked to our research questions. We used Stata, and user-written Stata commands for sequence analysis (SQ-Ados) originally developed by [105].

While the effectiveness and need for sequence analysis in social sciences has been questioned [106], it has remained a popular tool and has recently made significant advances [25]. At the forefront of such advances is an integration of network analysis with sequence analysis, such as in this study [107]. Sequence analysis has been used in various fields, including psychology [108, 109], sociology [102, 103, 110], economics [111, 112], history [104], public administration [113], and political science [114]. It has also been used to study housing careers in the US and European contexts, but, to the best of our knowledge, our paper presents the first application of sequence analysis to study housing conditions in the context of developing countries.

## Measurements for housing conditions

We measure housing conditions in terms of four aspects of housing: access to water, access to sanitation, durable structure, and form of tenure. We define the specific categories for levels of access to water and sanitation using service ladders proposed by the WHO and UNICEF Joint Monitoring Program (JMP) for Drinking Water [115] and Sanitation [116]. We define levels of access to water from lowest to highest levels of service: i) Level 0 when access to water is from an unimproved source, ii) Level 1 when access to water is from an improved source but is not on premises, and iii) Level 2 when access to water is from an improved source on premises. We define levels of access to sanitation from lowest to highest levels of service: i) Level 0 for open defecation, ii) Level 1 for an unimproved sanitation facility, iii) Level 2 for improved sanitation, and iv) Level 3 for a piped sewer on premise.

We created three categories to define levels of durability of structure: i) Level 0 for temporary structure, ii) Level 1 for semipermanent structure, and iii) Level 2 for permanent structure. We used building materials used for walls, roofs, and floors to create these categories. Houses made with durable wall materials such as bricks, durable roof materials such as cement concrete, and durable floor materials such as tiles were considered Level 2, whereas houses built with temporary materials for walls (e.g., bamboo and grass), roofs (e.g., grass, leaves, and bamboo), and floor (e.g., earth) were considered Level 0. Houses with walls built of durable materials but roofs and/or floors built of temporary materials were considered Level 1.

We used three categories to define forms of tenure, with an assumption that ownership is the highest form of secured tenure, followed by renting and free accommodation: i) Level 0 for individuals who lived for free, ii) Level 1 for individuals who rented a house, and iii) Level 2 for individuals who lived in a house either owned by themselves or their family members, including spouses and parents.

We then considered all these levels as housing states for analyses that are repeated for each of them separately: i) level of access to water, ii) level of access to sanitation, iii) durability of structure, and iv) form of tenure. We used these housing states to construct a sequence for each individual's housing career collected from her residential history in the survey. In that sense, if an individual moved twice and had Level 0 access to water in both homes, the sequence of her housing career with respect to water will be '0 0.' If this example individual had Level 0 access to sanitation in her first home, but the situation improved to Level 1 access to sanitation in her second home, then her housing career with respect to sanitation will be '0 1.'

We use the logic of the Slum Severity Index (SSI) originally developed by [99–101] that measures multiple housing deprivations in a single index to create the Housing Quality Index (HQI), which combines all four housing qualities, namely, access to water, access to sanitation, durable structure, and tenure, into a single score. Unlike [99–101], HQI uses multiple levels within individual indicators and is normalized to range between 0 and 1. Also, it is reciprocal with the SSI (i.e., it measures housing quality rather than housing deprivation), with 0 indicating the lowest housing quality and 1 indicating the best housing quality across all four dimensions. However, we should note that considering that all four dimensions are given equal weights in our index, it does not reflect the different values that residents may put on each dimension (e.g. residents might consider access to water more important than access to sanitation). This unavoidable limitation apart, we still think operationalizing a composite index is useful because a housing decision presents a bundled choice that may involve trade-offs between various housing characteristics (e.g., an individual may select her next home that improves access to water at the cost of lowered access to sanitation). Finally, the index incorporates time into the calculation to capture the entire lived history of an individual in a single score that ranges from 0 to 1.

$$HQI_i = \frac{\left(\frac{\sum x_{ijk} * t_{ij}}{k}\right)}{t_i}$$

where $HQI_i$ is the Housing Quality Index for an individual $i$;

 $X_{ijk}$ is the level of housing quality for individual $i$ at house $j$ on dimension $k$;

 $t_{ij}$ is the number of years individual $i$ spent at house $j$;

 $k$ is the number of dimensions; and

 $t_i$ is the total number of years of the entire residential history for individual $i$.

In essence, HQI captures the quality of housing on all four dimensions for the entire residential history in a single score that ranges between 0 and 1. Thereby, the index allow us to meaningfully compare individuals with each other regardless of the varying lengths of their residential histories involving the varying numbers of homes that they may have occupied.

## Data organization

We organized our data in three different ways to accomplish different types of analyses. First, we organized data as sequences of housing states for each of the dimensions to perform sequence analysis. Second, we organized data with a dyad of current and previous houses as units to record the housing state on each dimension in the present and previous houses. We used this data to calculate transition probabilities and perform analysis related to housing quality improvements with each residential move. Finally, we used individuals as units of analysis to perform analysis on HQI which is an individual level score.

## Results

We first present descriptive statistics from sequence analyses for water, sanitation, durable structure, and form of tenure. Since there are multiple sequences possible with varying numbers of moves and the possibility of repeating states, it is typical for sequence analysts to present the top *x* sequences as the most frequently occurring sequences. Unlike traditional descriptive statistics, we are interested in the most frequent occurrences as opposed to central tendencies. We combine these top sequences in analytically meaningful categories and present them as fewer categories within each indicator's context. Such grouping is our own and is subjective, but it allows us to observe the broader patterns in otherwise numerous sequence

possibilities. Next, we present the dynamics of housing conditions using sequence index plots and parallel coordinate plots, two of the conventional visualization techniques to present patterns in whole sequences. We present these results for the entire sample, followed by a comparison between migrants and non-migrants, as well as between shanty and non-shanty residents. Next, we present stochastic patterns between various housing states, followed by a section on temporal trends in the type of housing moves, an analysis of trade-offs between different housing conditions between moves, and a final section that analyzes the entire residential history of respondents using HQI.

## Patterns in housing quality

**Access to water.** It is evident from Table 1 that a high proportion of individuals (53%) have not experienced improvements in access to water despite residential mobility, as indicated by access to water remaining stable at either Level 0 or Level 1. In fact, individuals with Level 2 access are rare, and many with Level 2 access have experienced adverse outcomes with residential moves (as reflected in the sequences 2 1 and 2 1 1 that represent 11% of sampled individuals). Of particular interest are patterns where individuals with the lowest access did not experience any improvement in their access to water despite changing homes. Many individuals exhibit sequences such as 0, 0 0, and 0 0 0, indicating they were either not able to move out of deprived homes or did not see any improvements despite moving homes. We refer to these individuals as *perpetually deprived* and call for attention to them as target beneficiaries of water policies and programs. Similarly, individuals with Level 1 access have also often remained within that category, as indicated by the sequences 1, 1 1, 1 1 1, 1 1 1 1, and 1 1 1 1 1, all part of the top 20 sequences. This may be reflective of the fact that the overall housing stock in the city with higher levels of water access is in short supply, meaning this level of access is an

**Table 1. Top 20 sequence patterns and clusters for access to water.**

| Water (N = 1,063) | | | |
|---|---|---|---|
| **Type of Sequence** | | **Sequence** | **%** |
| Stable (53.3%) | Perpetually Deprived (7%) | 0 | 1.9 |
| | | 0 0 | 4.1 |
| | | 0 0 0 | 2 |
| | Stable at the Basic Level (40.8%) | 1 | 5.8 |
| | | 1 1 | 13.2 |
| | | 1 1 1 | 12.3 |
| | | 1 1 1 1 | 6.9 |
| | | 1 1 1 1 1 | 2.6 |
| | Privileged (4.5%) | 2 2 | 4.5 |
| Upward Mobility (30.2%) | Gentle Upward Mobility (28%) | 0 1 | 9.2 |
| | | 0 1 1 | 8.1 |
| | | 0 1 1 1 | 5 |
| | | 0 0 1 | 3.1 |
| | | 1 2 | 2.6 |
| | Leapfrogging (2.2%) | 0 2 | 2.2 |
| Downward Mobility (16.3%) | Downward to the Basic Level (11.1%) | 2 1 | 5.4 |
| | | 2 1 1 | 3.5 |
| | Downward to the Bottom Level (7.4%) | 1 0 | 5 |
| | | 1 1 0 | 2.4 |
| Oscillating (2.2%) | Upward with Setback (2.2%) | 1 2 1 | 2.2 |

indicator of the overall development level of the city. Individuals with Level 1 access do indicate a higher level of access than a complete lack of access (i.e., Level 0). Therefore, even if they have not experienced upward mobility over time, they are perhaps of less concern than those who stay at Level 0 for perpetuity.

Sequences such as 0 1, 0 2, and 1 2 are of particular interest because those individuals have experienced upward mobility in their access to water, with some individuals even skipping Level 1 by leapfrogging from 0 to 2. Similarly, individuals who started with Level 2 access and stayed at that level (2 2) are also of particular interest as a privileged group that begins with the best access to water possible and are able to maintain it.

The most critical sequences are the ones that show downward mobility, such as individuals represented by the sequence 1 0 (i.e., their first home had Level 1 access and the second home had Level 0 access). These individuals may have experienced a financial shock (such as losing employment), resulting in accepting inferior housing conditions. Even worse, they may be migrants who opted to move to inferior housing in the city to access other opportunities (such as the labor market) at the cost of better housing conditions in their rural origins. Such a group requires a very different kind of policy intervention because stability or improvement at any given level is essential and desirable policy goal. There are several groups who experience upward mobility only to return to the lower level where they started (i.e., the group with the sequence 1 2 1). Such oscillations are also of interest, since they might be portraying a very different kind of lived experience.

**Access to sanitation.** Housing careers display similar patterns when we look at improvements in access to sanitation (Table 2). Multiple individuals are *perpetually deprived* of sanitation (e.g., people with housing sequences 0 0, 0 0 0, and 0 0 0 0) that appear in the top 20 sequences. The pattern is similar for Level 1 access, which also represents unimproved sanitation (sequences 1 1, 1 1 1, 1 1 1 1, and 1 1 1 1 1).

**Table 2. Top 20 sequence patterns and clusters for access to sanitation.**

| Sanitation (N = 1,187) | | | |
|---|---|---|---|
| **Type of Sequence** | | **Sequence** | **%** |
| Stable (62.3%) | Perpetually Deprived (25.2%) | 0 | 3.3 |
| | | 0 0 | 11.7 |
| | | 0 0 0 | 8 |
| | | 0 0 0 0 | 2.2 |
| | Stable at the Basic Level (34.7%) | 1 | 4 |
| | | 1 1 | 11.4 |
| | | 1 1 1 | 11.2 |
| | | 1 1 1 1 | 6.2 |
| | | 1 1 1 1 1 | 1.9 |
| | Privileged (2.4%) | 2 2 | 2.4 |
| Upward Mobility (52.8%) | Gentle Upward Mobility (21.8%) | 0 1 | 9.2 |
| | | 0 1 1 | 7.3 |
| | | 0 1 1 1 | 5.3 |
| | Delayed Upward Mobility (7.1%) | 0 0 1 | 4.7 |
| | | 0 0 0 1 | 2.4 |
| | Upward Mobility to Privileged (22.1%) | 1 2 | 2.1 |
| | | 1 1 2 | 20 |
| | Leapfrogging (1.8%) | 0 2 | 1.8 |
| Downward Mobility (2.2%) | Downward to Bottom Level (2.2%) | 1 0 | 2.2 |
| Oscillating (2.9%) | Upward with Setback (2.9%) | 0 1 0 | 2.9 |

Similar to water, sanitation also displays patterns that experience upward mobility (e.g., 0 1, 0 1 1, 0 0 1, 0 1 1 1, and 1 2) and downward mobility (e.g., 1 0). It seems that there may be trade-offs involved if these are choices made by individuals (e.g., getting access to a better labor market at the cost of poor access to water and sanitation). It is worth noting that the highest level of access to sanitation does not appear in the top sequences which is in line with the context described earlier–only very small proportion of people have access to the highest level of sanitation in recent times and this context was not better in the past.

**Durable structure.**   A durable structure, as reflected in the materials used for walls, roofs, and floors, shows that a vast majority of individuals (77.4%) in Dar es Salaam had stable housing patterns with no temporary structures in their history (Table 3). Unlike water and sanitation, the lowest level of housing condition in terms of durable structure (Level 0) is not part of the lived histories of our sampled residents. However, only a small minority had housing of a permanent nature (Level 2), and many of them experienced a downward trajectory, thus falling back to the semipermanent category (Level 1). Upward mobility is also on the lower side (20.2%) compared to that observed for water and sanitation. This finding is in line with the fact that the opportunity structure that the housing market in Dar es Salaam has presented in the last several decades is reasonably stable for housing stock of a semipermanent nature, as discussed in the case study context.

**Form of tenure.**   Form of tenure as recorded at the three levels, free (Level 0), rented (Level 1), and owned (Level 2), also shows that downward mobility is much higher (54.9%) compared to the other housing conditions (Table 4). Specifically, owners are falling back to become renters and staying renters in subsequent moves. The most common reasons for moves that accepted a lower level form of tenure included separation from parental homes upon adulthood, housing moves triggered by family events, the need for a larger home, the

**Table 3. Top 20 sequence patterns and clusters for durable structure.**

| Durable Structure (N = 1,678) | | | |
|---|---|---|---|
| Type of Sequence | | Sequence | % |
| Stable (77.8%) | Stable at the Basic Level (77.4%) | 1 | 9.12 |
| | | 1 1 | 22.7 |
| | | 1 1 1 | 22.8 |
| | | 1 1 1 1 | 14.6 |
| | | 1 1 1 1 1 | 5.5 |
| | | 1 1 1 1 1 1 | 1.9 |
| | | 1 1 1 1 1 1 1 | 1.0 |
| | Privileged (0.4) | 2 2 | 0.4 |
| Upward Mobility (20.2%) | Gentle Upward Mobility (17%) | 0 1 | 5.2 |
| | | 0 1 1 | 4.7 |
| | | 0 1 1 1 | 4.4 |
| | | 0 1 1 1 1 | 1.9 |
| | | 0 1 1 1 1 1 | 0.9 |
| | Delayed Upward Mobility (1.5%) | 0 0 1 | 0.8 |
| | | 0 0 1 1 | 0.8 |
| | Upward Mobility to Privileged (1.6%) | 1 2 | 1.0 |
| | | 1 1 2 | 0.7 |
| Downward Mobility (1.1%) | Downward to Bottom Level (1.1%) | 2 1 | 0.7 |
| | | 2 1 1 | 0.5 |
| Oscillating (0.8%) | Upward with Setback (0.8%) | 1 2 1 | 0.8 |

**Table 4. Top 20 sequence patterns and clusters for the form of tenure.**

| Form of Tenure (N = 1365) | | | |
|---|---|---|---|
| Type of Sequence | | Sequence | % |
| Stable (27.9%) | Perpetually Deprived (1.9%) | 0 | 1.9 |
| | Stable at the Basic Level (4.0%) | 1 1 | 1.8 |
| | | 1 1 1 | 2.2 |
| | Privileged (22%) | 2 | 3.7 |
| | | 2 2 | 11.2 |
| | | 2 2 2 | 5.4 |
| | | 2 2 2 2 | 1.7 |
| Upward Mobility (1.8%) | Upward Mobility to Privilege (1.8%) | 0 2 | 1.8 |
| Downward Mobility (54.9%) | Quick Downward Mobility (41.8%) | 2 1 | 17.6 |
| | | 2 1 1 | 12.0 |
| | | 2 1 1 1 | 7.7 |
| | | 2 1 1 1 1 | 2.6 |
| | Delayed Downward Mobility (13.2%) | 2 2 1 | 6.4 |
| | | 2 2 1 1 | 5.2 |
| | | 2 2 1 1 1 | 1.6 |
| | Downward to Bottom Level (3.1%) | 2 0 | 3.1 |
| Oscillating (14.1%) | Downward Mobility and Recovery (9.4%) | 2 1 2 | 7.7 |
| | | 2 1 2 2 | 1.7 |
| | Downward Mobility and Slow Recovery (4.7%) | 2 1 1 2 | 2.9 |
| | | 2 1 1 1 2 | 1.8 |

need for a cheaper home, and the need for a safer neighborhood. However, it is worth noting that 22% of owners remained owners despite several moves while citing similar reasons for their moves.

## Dynamics of changes in housing conditions

Overall dynamics of housing conditions are best captured for individual households with sequence index plots and parallel-coordinate plots. Unlike the top sequences, these plots include all the households and provide an overall picture of residential histories, including the number of housing moves. Sequence index plots show individuals' housing move trajectories as horizontal lines, which are grouped by similarity in initial conditions and bifurcated by differences in subsequent conditions. The parallel-coordinate plot indicates the volume of variety of housing career paths, where line thickness indicates the proportion of individuals on that path, as a different way to visualize the housing career information.

**Dynamics in access to water.** There is a large segment of the population that starts its housing career with Level 0 access to water (coded in blue in Fig 1A). Many of them do not experience upward mobility even as they change their first homes (continued as blue with housing moves presented on the x-axis). Some of them do not experience any upward mobility despite changing eight or more homes.

Approximately one-sixth of individuals started their careers at the highest access level (coded green) and were able to maintain it through their residential moves. Some of them lost their access (green changing to magenta in the sequence index plot of Fig 1A). These individuals have experienced downward mobility as they changed homes (the same holds true for magenta changing into blue at multiple points). Particularly vulnerable populations that

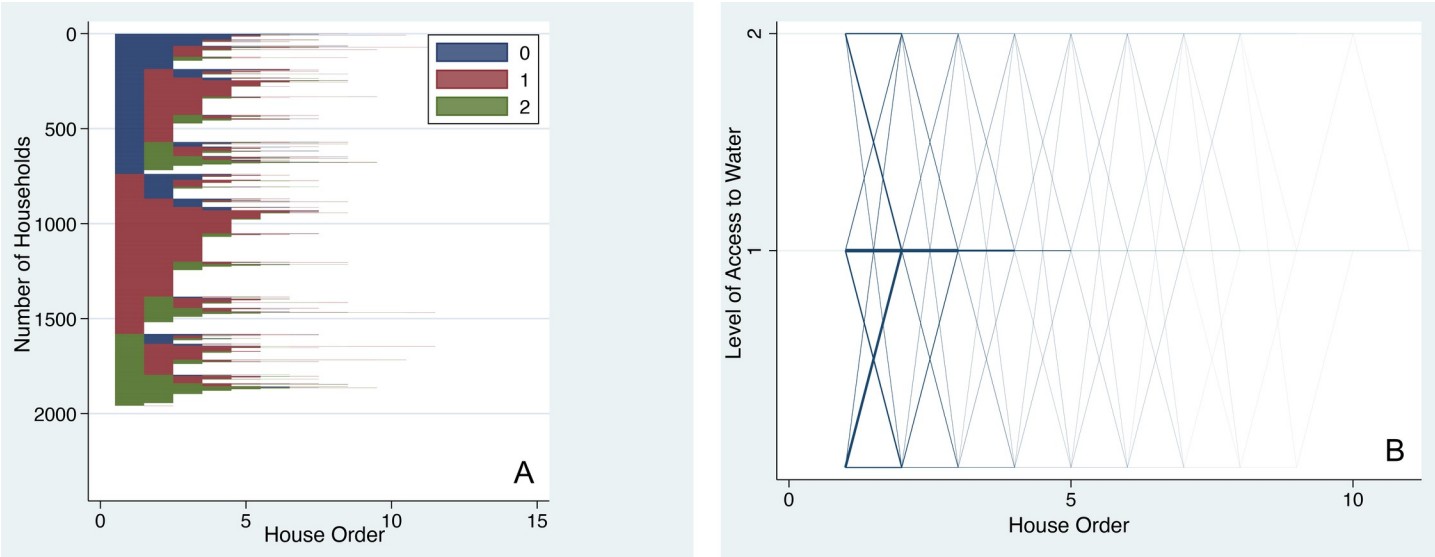

**Fig 1. Dynamics of access to water with residential moves.** (A) Sequence index plot of access levels for water shows individual housing trajectories as horizontal lines grouped by initial conditions. (B) Parallel-coordinate plot of access levels for water: line thickness shows volume of flow for a given housing trajectory.

experience downward mobility should be of interest to policymakers. Our analysis is a first step in that direction.

**Dynamics in access to sanitation.** Housing careers in sanitation show an even more dire picture, with many individuals starting with homes where open defecation is the only option and the conditions have not changed for them despite multiple housing moves. While many individuals eventually experience upward mobility, it is concerning that such a high proportion begin their careers with poor housing when it comes to sanitation (Fig 2). It is not surprising that almost none of the individuals start at the highest level of access to sanitation (i.e.,

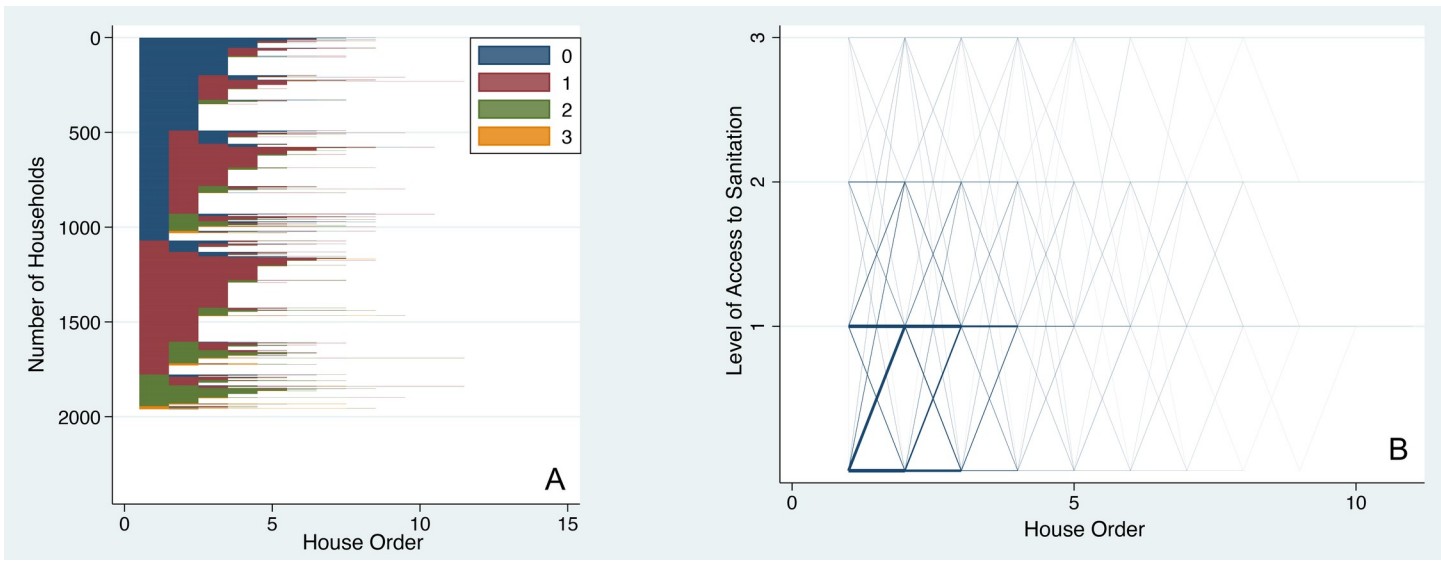

**Fig 2. Dynamics of access to sanitation with residential moves.** (A) Sequence index plot of access levels for sanitation shows individual housing trajectories as horizontal lines grouped by initial conditions. (B) Parallel-coordinate plot of access levels of sanitation: line thickness shows volume of flow for a given housing trajectory.

toilet piped to a sewer, or Level 3 as indicated by the color yellow) given that a very small proportion of housing stock in Dar es Salaam provides such an access, thus limiting the opportunity. A large proportion of people starts at Level 1 and moves to Level 2 access. The parallel-coordinate plot (Fig 2B) shows the same information in an alternate form, where density of households experiencing a similar housing career is shown with varying line thicknesses.

**Dynamics in houses with durable structure.** Housing careers with regard to durable structure show that a large proportion of residents start their career with semipermanent housing and a very small fraction of them are able to move to housing with permanent structures with brick walls, cement concrete roofs and tiled floors. Given the housing context of Dar es Salaam of the time in which most residents have lived, this is not surprising. A significantly large proportion of residents began their housing career with temporary structures (Blue in Fig 3A) but moved to semipermanent structures as soon as their second houses (Magenta). The oscillation between various types of housing structure is less common, as seen in Fig 3B, and hence the city provides a stable housing environment, even if it is far from the desirable urban planning standards for permanent structures.

**Dynamics in the form of tenure.** In terms of tenure, it is shocking that many residents begin their housing careers with owned homes and move into rental houses as soon as their second homes (Fig 4). The most cited reasons for such a move are: residents' separation from their parents at adulthood, and a need for cheaper housing. For those who begin their career as renters or become renters in their second house, a large proportion of them remain renters through at least the halfway point of their housing careers.

## Housing careers of migrants and non-migrants

We conducted a similar analysis for water, sanitation, durable structures, and forms of tenure for migrants and non-migrants separately to see if there are systematic differences in how both groups experience improvements in housing conditions. About 78% of the histories in our sample were for migrants and 22% were for non-migrants. The number of housing moves (~3) and average duration of stay (~7 years) for both groups were similar, with no statistically

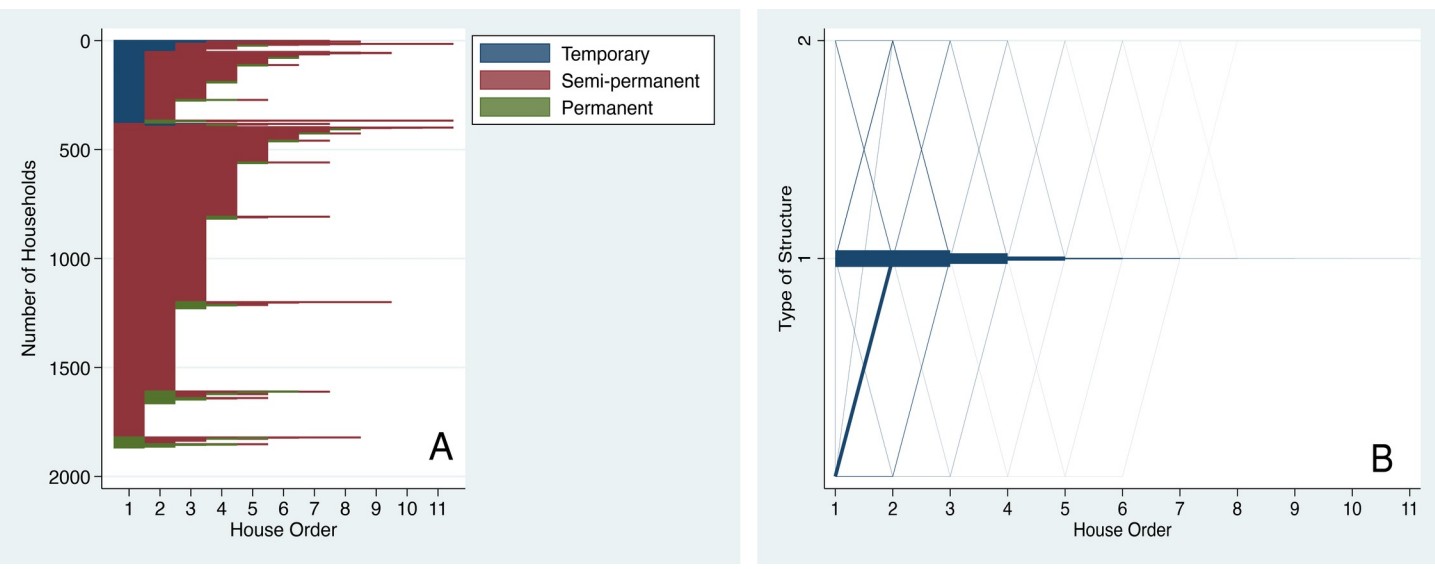

**Fig 3. Dynamics of housing with durable structure with residential moves.** (A) Sequence index plot for durable structure shows individual housing trajectories as horizontal lines grouped by initial conditions. (B) Parallel-coordinate plot for durable structure: line thickness shows volume of flow for a given trajectory.

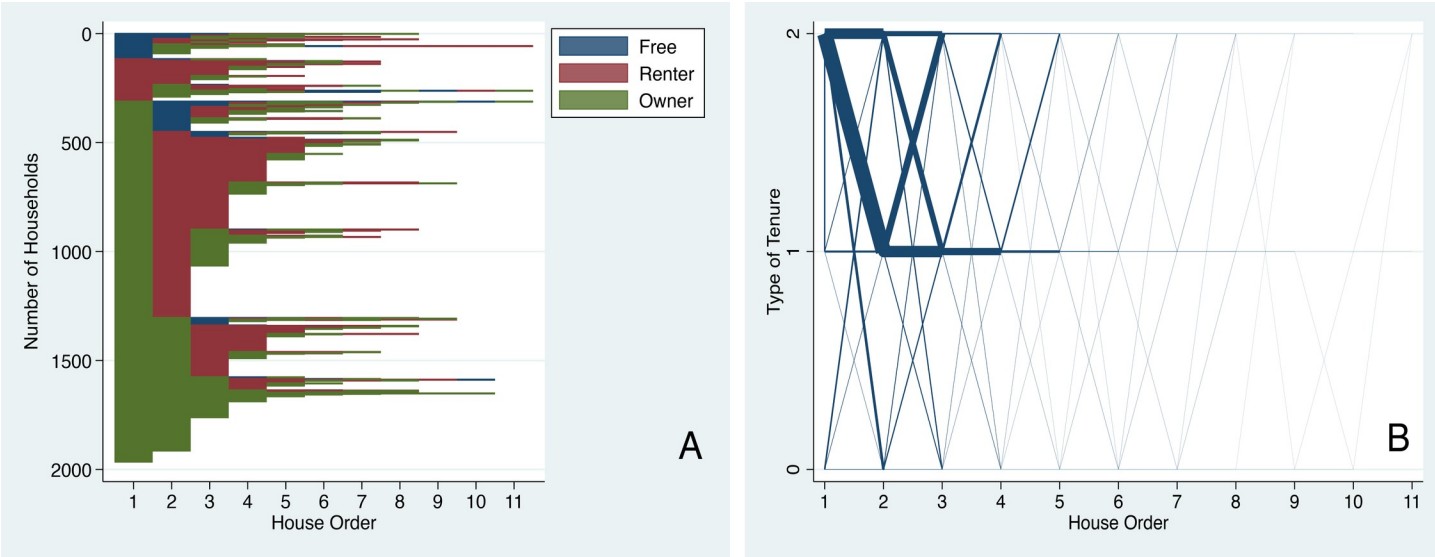

**Fig 4. Dynamics of form of tenure with residential moves.** (A) Sequence index plot for tenure forms shows individual housing trajectories as horizontal lines grouped by initial conditions. (B) Parallel-coordinate plot for tenure forms: line thickness shows volume of flow for a given trajectory.

significant differences. It is evident from Table 5 that a higher proportion of non-migrants have stability in their access to water compared to migrants. It is largely because a much higher proportion of migrants experience upward mobility as they move to the city. While the trajectory remains similar after arrive to the city as seen in S1.1 Fig in S1 File, migrants experience poor amenities in the early part of their lives in their origins and see improvements only after arriving in the city. The difference in proportions experiencing downward mobility and oscillations are small with slightly higher proportions for non-migrants.

A similar gap is clearly evident when it comes to access to sanitation. The residential histories 0 0, 0 0 0, and 0 0 0 0 are experienced by much higher proportions of migrants (23.1%) than non-migrants (12.3%) in terms of access to sanitation, suggesting higher perpetual deprivation among migrants. A visible difference is evident in S1.2 Fig in S1 File, where we compare all migrants with all non-migrants using sequence index plots and parallel-coordinate plots. A significant proportion of migrants start their housing careers with Level 0 access to sanitation in their origins compared to those who are non-migrants born in homes with better sanitation conditions in Dar. However, both groups have similar proportion of perpetually deprived residents as seen in Table 6. A much higher proportion of non-migrants experience stability

**Table 5. Patterns and clusters of sequences in access to water for migrants and non-migrants.**

| Types of Sequences | | Migrants | | Non-migrants | |
|---|---|---|---|---|---|
| Stable | Perpetually Deprived | 42.7 | 6.7 | 65.8 | 8.4 |
| | Stable at the Basic Level | | 31.1 | | 51.6 |
| | Privileged | | 4.9 | | 5.8 |
| Upward Mobile | Gentle Upward Mobile | 28 | 25.1 | 6.3 | 6.3 |
| | Leapfrogger | | 2.9 | | 0 |
| Downward Mobile | Downward to the Basic Level | 17 | 10.2 | 19.5 | 10.6 |
| | Downward to the Bottom Level | | 6.8 | | 8.9 |
| Oscillating | Upward with Setback | 2.1 | 2.1 | 2.9 | 1.7 |
| | Downward and Recovered | | 0 | | 1.2 |

**Table 6. Patterns and clusters of sequences in access to sanitation for migrants and non-migrants.**

| Types of Sequences | | Migrants | | Non-migrants | |
| --- | --- | --- | --- | --- | --- |
| Stable | Perpetually Deprived | 45.6 | 23.3 | 75.6 | 22.5 |
| | Stable at the Basic Level | | 19.9 | | 47 |
| | Privileged | | 2.4 | | 6.1 |
| Upward Mobile | Gentle Upward Mobile | 42.8 | 25.5 | 15.7 | 11.1 |
| | Delayed Upward Mobile | | 11.6 | | 2.6 |
| | Upward Mobile to Privileged | | 3.7 | | 2 |
| | Leapfrogger | | 2 | | 0 |
| Downward Mobile | Downward to the Basic Level | 0 | 0 | 8.9 | 2.7 |
| | Downward to the Bottom Level | | 0 | | 6.2 |
| Oscillating | Upward with Setback | 3.3 | 3.3 | 0 | 0 |

compared to migrants, partly because a higher proportion of migrants experience upward mobility. It is noteworthy that non-migrants experience higher downward mobility compared to migrants partly because they start at a privileged position (often in their parents' homes) and move to lower level as they form their own household.

There is a clear difference that is evident between migrants and non-migrants when it comes to a durable structure (Table 7). This may be because migrants' housing in origin (often in rural areas) may not be made of permanent materials and hence their first house in the city (2nd house in S1.3A Fig in S1 File) is an upward trajectory with semipermanent housing. A slightly higher proportion of Non-migrants have stability in type of structures they have occupied with most of them occupying Level 1 structures (semipermanent).

Similar to the structure, migrants show high levels of ownership for their homes in rural origins, but a much larger proportion begin their career in the city with rental housing (S1.4A Fig in S1 File 2nd house order in Magenta). Such decline in the form of tenure is visible in non-migrants as well (S1.4B and S1.4D Fig in S1 File) but is more common among migrants. As showed in Table 8, non-migrants have stable forms of tenure with more than third maintaining ownership. Ownership remains elusive to migrants and 58% of them experience downward mobility with many of them experiencing it as soon as they move to the city.

## Housing careers in shanty and non-shanty neighborhoods

We performed a similar analysis of access to water, sanitation, durable structures, and tenure for shanty and non-shanty neighborhoods to explore whether those who currently live in shanty neighborhoods have particularly experienced housing deprivations in their life-courses

**Table 7. Patterns and clusters of sequences in type of structure for migrants and non-migrants.**

| Type of Sequence | | Migrants | | Non-migrants | |
| --- | --- | --- | --- | --- | --- |
| Stable | Perpetually Deprived | 75.3 | 0.4 | 89.7 | 0 |
| | Stable at the Basic Level | | 74.9 | | 88.4 |
| | Privileged | | 0 | | 1.3 |
| Upward Mobile | Gentle Upward Mobile | 25.3 | 21.5 | 6.6 | 3.9 |
| | Delayed Upward Mobile | | 2 | | 0.2 |
| | Upward Mobile to Privileged | | 1.3 | | 2.5 |
| | Leapfrogger | | 0.5 | | 0 |
| Downward Mobile | Downward to the Basic Level | 0.5 | 0.5 | 2.3 | 2.3 |
| Oscillating | Upward with Setback | 1 | 1 | 1.3 | 1.3 |

**Table 8. Patterns and clusters of sequences in forms of tenure for migrants and non-migrants.**

| Type of Sequence | | Migrants | | Non-migrants | |
|---|---|---|---|---|---|
| Stable | Perpetually Deprived | 18.9 | - | 51.4 | 6.8 |
| | Stable at the Basic Level | | 3.3 | | 9.4 |
| | Privileged | | 15.6 | | 35.2 |
| Upward Mobile | Gentle Upward Mobile | 0 | 0 | 4.2 | 1.1 |
| | Upward Mobile to Privileged | | 0 | | 3.1 |
| Downward Mobile | Quick Downward Mobile | 57.9 | 41.6 | 38.3 | 25.9 |
| | Delayed Downward Mobile | | 12 | | 7.6 |
| | Downward to Bottom Level | | 2.7 | | 4.8 |
| | Downward to Bottom Level and Recovering | | 1.6 | | 0 |
| Oscillating | Upward Mobile with Setback | 17.8 | | 4.8 | 2.5 |
| | Downward Mobile and Recovered | | 12.4 | | 2.3 |
| | Downward Mobile and Slowly Recovered | | 5.4 | | 0 |

that are different from those who currently live in non-shanty neighborhoods. It is evident from Table 9 that differences between shanty and non-shanty residents are negligible. S2.1 Fig in S2 File presents these patterns visually.

A similar pattern is observed in access to sanitation where residential histories are similar between both the groups (Table 10). A slightly higher proportion of non-shanty neighborhood residents (24.8%) are perpetually deprived (0, 0 0, 0 0 0, and 0 0 0 0) than those in shanty neighborhoods (21%). However, a smaller proportion of shanty residents have observed upward mobility compared to non-shanty residents. S2.2 Fig in S2 File presents these patterns visually.

We do not observe distinctly different patterns between residents of shanty neighborhoods and non-shanty neighborhoods when it comes to the durability of the structure (Table 11). Both groups seem to have a similar experience with one exception: a large proportion of shanty residents have been in semipermanent structures (78.9%) despite multiple housing moves. This proportion is 62.8% for non-shanty residents. S2.3 Fig in S2 File presents these patterns visually.

In both groups, owners ending up in rental housing is fairly common (Table 12). We do not see distinctly different patterns between shanty and non-shanty residents with one notable exception: a much higher proportion of shanty residents have seen downward mobility (69.3%) compared to non-shanty residents (56.2%). S2.4 Fig in S2 File presents these patterns visually.

**Table 9. Patterns and clusters of sequences in access to water for shanty and non-shanty residents.**

| Types of Sequences | | Shanty | | Non-shanty | |
|---|---|---|---|---|---|
| Stable | Perpetually Deprived | 55.8 | 5.3 | 49.6 | 7.4 |
| | Stable at the Basic Level | | 46.8 | | 38.0 |
| | Privileged | | 3.7 | | 4.2 |
| Upward Mobile | Gentle Upward Mobile | 29.0 | 26.7 | 28.1 | 26.1 |
| | Leapfrogger | | 2.3 | | 2.0 |
| Downward Mobile | Downward to the Basic Level | 13.6 | 10.6 | 15.1 | 8.2 |
| | Downward to the Bottom Level | | 3.0 | | 6.8 |
| Oscillating | Upward with Setback | 0.0 | 1.6 | 1.6 | 2.0 |

**Table 10. Patterns and clusters of sequences in access to sanitation for shanty and non-shanty residents.**

| Types of Sequences | | Shanty | | Non-shanty | |
|---|---|---|---|---|---|
| Stable | Perpetually Deprived | 61.7 | 21.0 | 61.2 | 24.8 |
| | Stable at the Basic Level | | 39.4 | | 34.1 |
| | Privileged | | 1.3 | | 2.3 |
| Upward Mobile | Gentle Upward Mobile | 28.4 | 21.2 | 33.8 | 21.4 |
| | Delayed Upward Mobile | | 5.9 | | 7.0 |
| | Upward Mobile to Privileged | | 1.3 | | 3.7 |
| | Leapfrogger | | 0.0 | | 1.7 |
| Downward Mobile | Downward to the Bottom Level | 1.3 | 1.3 | 2.2 | 2.2 |
| Oscillating | Upward with Setback | 3.0 | 3.0 | 2.8 | 2.8 |

## Stochastic patterns in housing states

In this section, we present transition probabilities between various housing states for all four dimensions. The primary assumption in sequence analysis is that housing states do not appear at random, so one of the goals of sequence analysis is to detect patterns in the order of the housing states [25]. Such patterns could help in predicting which housing state might appear in the next housing move for an individual. Transition matrices and probability bubble graphs are conventional methods of presenting such stochastic patterns.

As is evident from Fig 5, deprived households in almost all four dimensions remain deprived in their next houses (e.g., 32% of movers who had Level 0 access to water remain at Level 0 in their next houses, as seen in Fig 5A). It is also clear that a large portion of privileged groups are able to maintain their privilege in the next houses (e.g., 46% movers who had Level 2 access to water find the same level of access to water in their next houses, as seen in Fig 5A). We also see larger probabilities for upward mobility (e.g., residents with Level 0 access to water had a 49% chance of having Level 1 access to water in their next houses).

## Temporal trends in housing improvements

We take two consecutive homes occupied by a resident as a dyad to study the patterns in housing improvements. Such an approach allows us to map these experienced improvements over time periods. As shown in Fig 6, there is a significant decline in residents who experienced an improvement in their level of access to water in the 2000s (22.2%) and 2010s (18.8%) compared to housing moves of the past (e.g., 33.3% moves brought improvement for individuals in 1960s). These histories are primarily a result of contextual factors that shape the housing environment of those periods, as discussed in the earlier section on the background of Dar es Salaam.

**Table 11. Patterns and clusters of sequences in type of structure for shanty and non-shanty residents.**

| | | Shanty | | Non-shanty | |
|---|---|---|---|---|---|
| Stable | Perpetually Deprived | 78.9 | 0.3 | 63.2 | 0.0 |
| | Stable at the Basic Level | | 78.6 | | 62.8 |
| | Privileged | | 0.0 | | 0.4 |
| Upward Mobile | Gentle Upward Mobile | 19.4 | 15.3 | 21.1 | 17.0 |
| | Delayed Upward Mobile | | 1.9 | | 2.5 |
| | Upward Mobile to Privileged | | 2.2 | | 1.6 |
| Downward Mobile | Downward to the Basic Level | 0.7 | 0.7 | 1.1 | 1.1 |
| Oscillating | Upward with Setback | 0.5 | 0.5 | 0.8 | 0.8 |

**Table 12. Patterns and clusters of sequences in forms of tenure for shanty and non-shanty residents.**

| Type of Sequence | | Shanty | | Non-shanty | |
|---|---|---|---|---|---|
| Stable | Perpetually Deprived | 24.6 | 2.2 | 26.2 | 1.9 |
| | Stable at the Basic Level | | 7.2 | | 4.0 |
| | Privileged | | 15.2 | | 20.3 |
| Upward Mobile | Upward Mobile to Privilege | 0.0 | 0.0 | 1.8 | 1.8 |
| Downward Mobile | Quick Downward Mobile | 69.3 | 48.2 | 56.2 | 39.9 |
| | Delayed Downward Mobile | | 16.5 | | 13.2 |
| | Downward to Bottom Level | | 3.1 | | 3.1 |
| | Downward to Bottom Level and Recovering | | 1.5 | | 0.0 |
| Oscillating | Downward Mobile and Recovered | 5.0 | 3.5 | 14.1 | 9.4 |
| | Downward Mobile and Slowly Recovered | | 1.5 | | 4.7 |

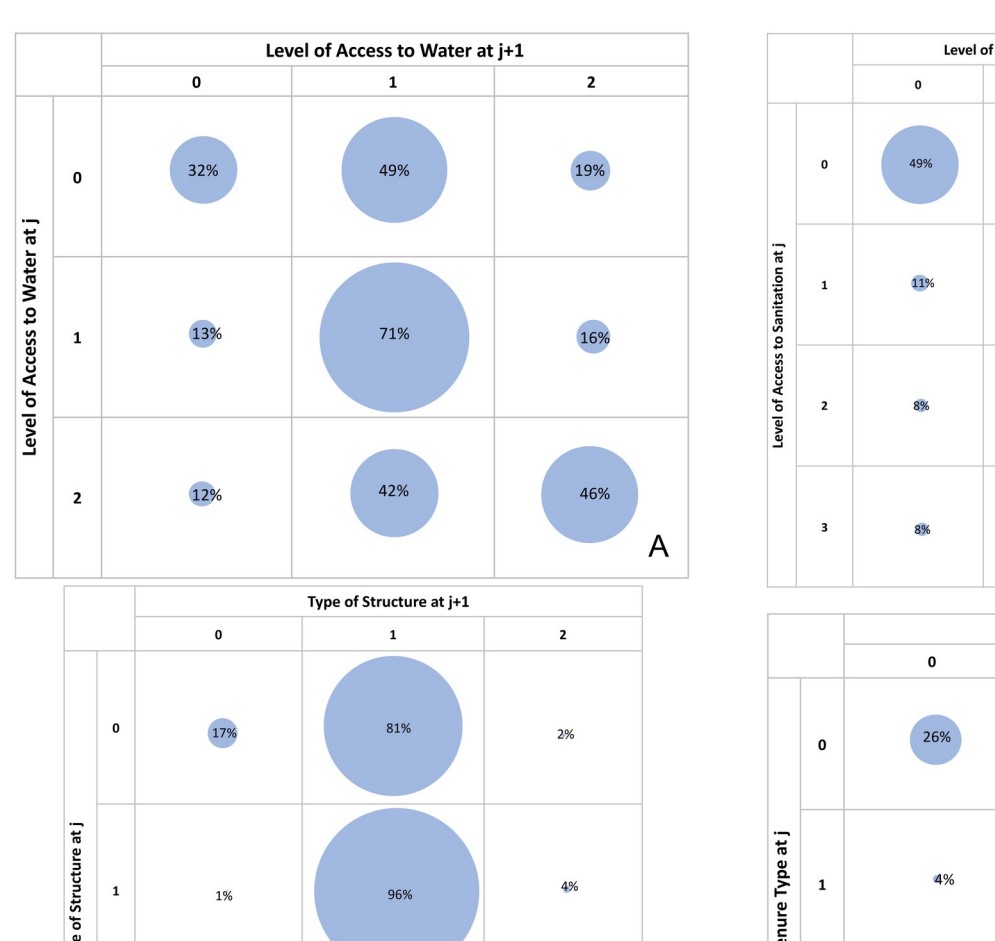

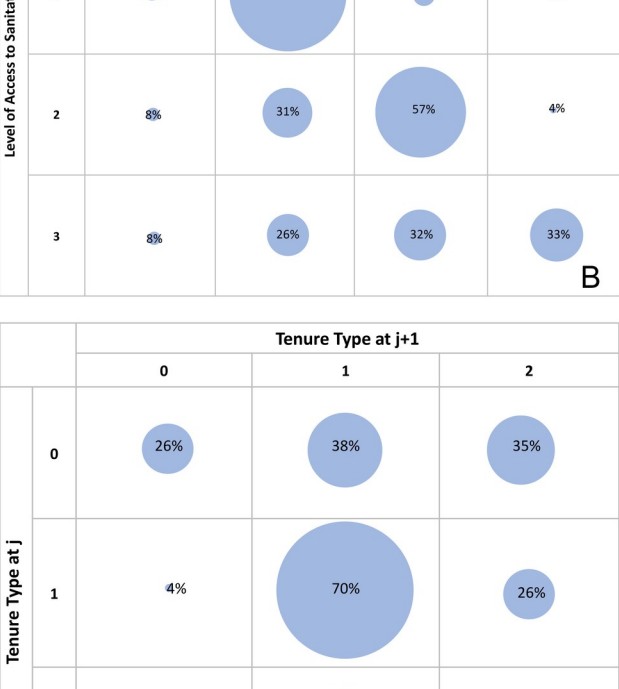

**Fig 5. Transition bubble graphs showing probabilities of transitions between housing states resulting from housing moves.** (A) Transition probabilities between levels of access to water. (B) Transition probabilities between levels of access to sanitation. (C) Transition probabilities between types of durable structure. (D) Transition probabilities between forms of tenure.

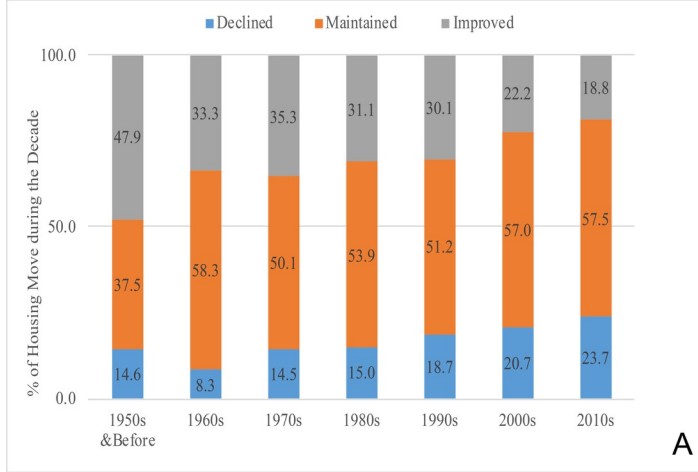

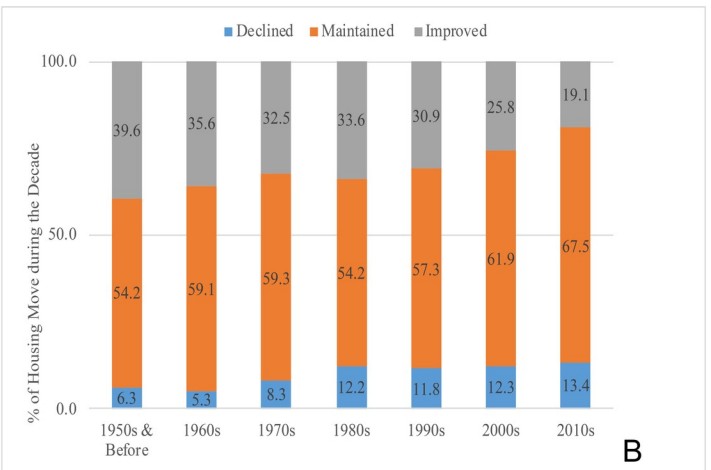

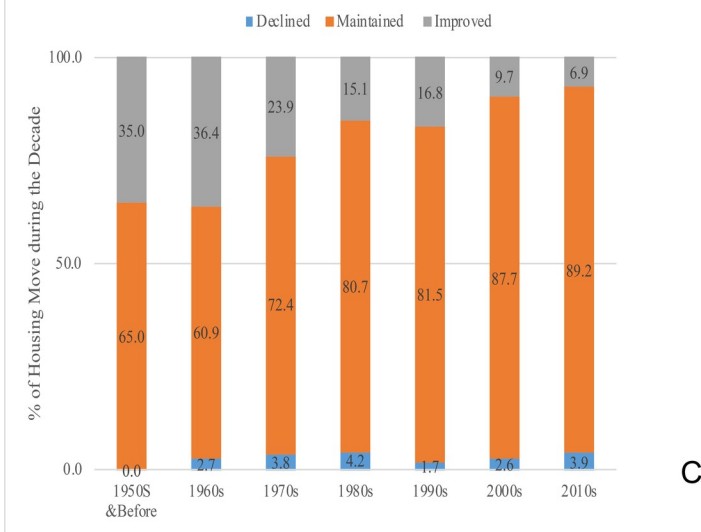

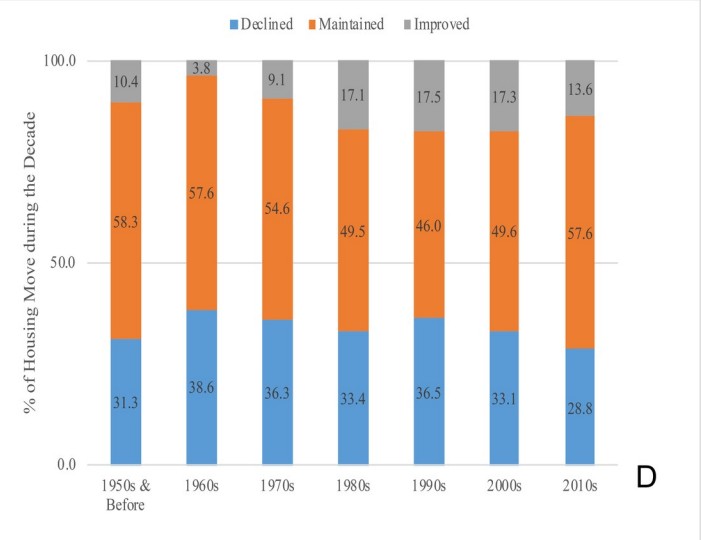

**Fig 6. Changes in housing quality for residential moves by periods (decades).** (A) Change in access to water with housing moves. (B) Change in access to sanitation with housing moves. (C) Change in type of structure with housing moves. (D) Change in forms of tenure with housing moves.

### Trade-offs between dimensions of housing

As [19] has shown, individuals accept trade-offs between housing conditions. For example, to own a house, individuals may move further away from their job location or accept a housing location where such basic services as water and sanitation are not yet available. This pattern is observed in Dar es Salaam among migrants located in peripheral neighborhoods [19]. We find evidence for such trade-offs by conducting a bivariate analysis of changes in housing quality in two dimensions at a time. For example, 16% of residential moves in our sample involved declines in access to water while observing improvements in access to sanitation. Similarly, 14% of residential moves involved declines in levels of access to sanitation while observing improvements in levels of access to water. Such analysis lends support to the fact that individuals have varied preferences, which the life-course approach captures well. Similarly, 17% of housing moves that improved the form of tenure (i.e., renter to owner) came at the cost of declined levels of access to sanitation. For a dynamic visualization of such trade-offs, we invite

readers to explore supplemental multimedia files S3 and S4 Files, which were created using Gapminder tools [117].

## Housing Quality Index: A snapshot of the entire residential history

As discussed in the methodology section, we calculated a single score that captures an individual's entire residential history that measures housing quality across all four dimensions and incorporates time (e.g., if person A lived in a deprived house with Level 0 access to water for a longer duration compared to person B who lived in the same condition for a shorter duration, then person A will have a lower score). Lower average of all residents (0.49) suggests that Dar es Salaam residents have lived in lower than optimal housing conditions over their life-course. Similarly, a high standard deviation (0.14) indicates heterogeneity among residents' lived experiences in Dar es Salaam as far as the quality of housing is concerned (Fig 7). While there was a statistically significant difference (at p < 0.01) between shanty and non-shanty residents, the difference was negligible (average HQI was 0.49 and 0.51 for shanty and non-shanty residents respectively). Similarly, there was no statistically significant difference between migrants and

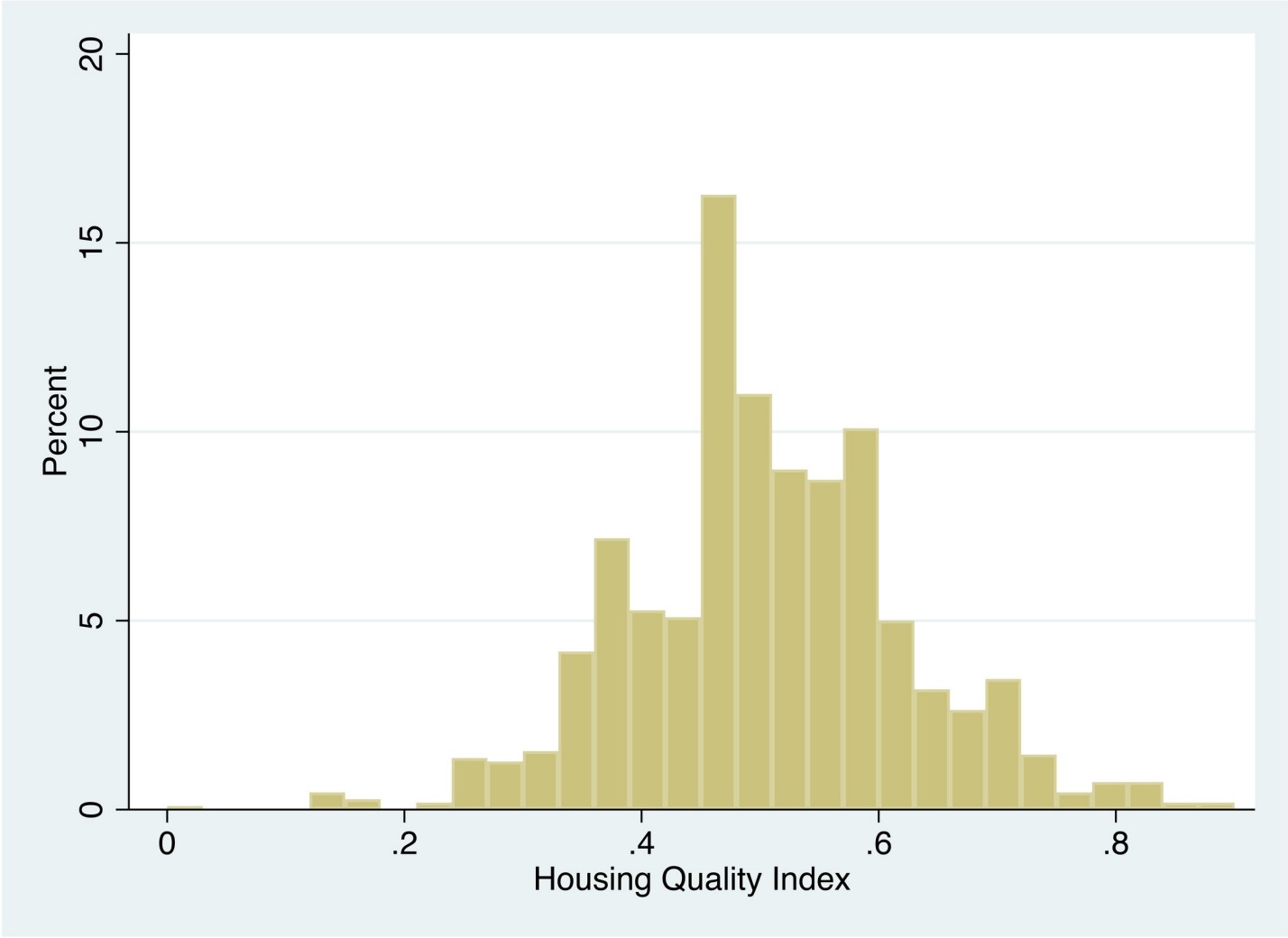

**Fig 7. Heterogeneity of Housing Quality Index across residents.**

non-migrants when we consider the entire residential history as reflected in average HQI of both the groups. Such analysis could be useful in asking questions such as who experiences better (or worse) housing quality over their life-course and what are the determinants of such differences in their housing experience. We hope that our analysis will precede such inquiries, which can be answered with rich interviews and focus group discussions, as done in the study [19].

## Discussion

The analyses in this paper confirm that housing careers do not necessarily show upward mobility in the developing world in general and for marginalized populations such as migrants and shanty residents in particular. For a significant proportion of individuals, moving homes does not lead to improvements in living conditions. In fact, several of the frequently occurring housing careers show a downward trend or stagnation. This confirms our hypothesis that residential mobility does not necessarily mean upward moves on the housing ladder. Many housing careers begin at the lowest level and remain that way for individuals regardless of their migration status or neighborhood characteristics.

While our study only focuses on one specific city and relies on a relatively small sample, it lends support to the almost nonexistent literature on mobility patterns in the global South. It should be considered an exploration in the rather unexplored territory of housing careers in the developing world. We hope that future studies find more positive results, as it seems that experiencing upward mobility is out of reach for many residents, including for many migrants and residents of shanty neighborhoods, despite moving homes multiple times.

We also demonstrated that there are trade-offs between various dimensions of housing. Residents trade improvement in one housing dimension (e.g. ownership) with decline in the other (e.g. sanitation). It should be noted that we observed only four parameters of housing, which do not provide us an opportunity to see trade-offs between other parameters of housing choice, such as dwelling size, type of neighbors, and location quality [17, 19].

Nonetheless, the approach of this paper to analyze housing careers over an entire residential history rather than a single residential move provides us with nuanced and meaningful evidence from a policymaking standpoint. Many individuals have lived in their present state for many years and will probably never reach the top of the ladder in their housing careers. Consequently, policies and programs that target improved housing and such basic services as water and sanitation could focus on residents that are experiencing deprivation over time rather than only across space. Currently deprived populations are known to exist in deprived states for long times, but evidence from a systematic analysis of housing careers, as demonstrated in this paper, brings this point forward much more convincingly than studies that focus on cross-sectional data or single residential moves alone.

We demonstrate that rural-to-urban migration leads to residential improvement for many migrants who are deprived in their rural origins, but eventual moves within the city do not bring further improvements for migrants in the same way that they do for non-migrants. However, in long run both migrants and non-migrants show no difference in their housing conditions as captured by a single normalized score. Similarly, moves within a city are not associated with upward mobility in housing conditions for shanty and non-shanty residents. Further investigation is required to understand whether individual residential moves, and especially those involving downward trajectories, are a result of forced evictions or involuntary moves associated with life events such as loss of employment or other financial catastrophes. In addition, recent advances in the integration of network analysis and sequence analysis to identify sources of connectedness between sequences through lived experiences (i.e., people with similar residential histories) as opposed to established social categories such as migrants,

or groupings based on only spatial proximity (e.g., people living in the same neighborhood such as a specific shanty) could advance our understanding of new kinds of networks that have not been observed previously [107]. Our dataset was limited in exploring such networks and causal mechanisms to explain variation in housing conditions. However, we believe that we have opened up a new area of investigation for housing policies in the developing world.

## Supporting information

**S1 File.**
(DOCX)

**S2 File.**
(DOCX)

**S3 File.**
(MP4)

**S4 File.**
(MP4)

## Acknowledgments

We are thankful to the World Bank Group's Spatial Development of Cities Program, which made the data available for this research. We are also thankful to Anne Shreshtha from the World Bank Group and Tanushree Bhan from the University of Massachusetts Boston for providing valuable feedback on earlier drafts of this manuscript. We are sincerely grateful to the academic editor and both the reviewers of this manuscript, who provided very constructive feedback on earlier versions.

## Author Contributions

**Conceptualization:** Amit Patel, George Joseph.

**Data curation:** Amit Patel, Namesh Killemsetty, Sokha Eng.

**Formal analysis:** Amit Patel, George Joseph, Namesh Killemsetty, Sokha Eng.

**Funding acquisition:** Amit Patel.

**Methodology:** Amit Patel, George Joseph.

**Project administration:** Amit Patel.

**Software:** Amit Patel, Sokha Eng.

**Supervision:** Amit Patel, George Joseph.

**Validation:** Amit Patel.

**Visualization:** Amit Patel, Sokha Eng.

**Writing – original draft:** Amit Patel, Namesh Killemsetty.

**Writing – review & editing:** Amit Patel, George Joseph, Namesh Killemsetty, Sokha Eng.

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
