## [Decision Letter · Decision Letter 0]

27 May 2020

PONE-D-20-11700

Effects of Residential Mobility and Migration on Standards of Living in Dar es Salaam, Tanzania: A Life-course Approach

PLOS ONE

Dear Dr. Patel,

Thank you for submitting your manuscript to PLOS ONE. After careful consideration, we feel that it has merit but does not fully meet PLOS ONE’s publication criteria as it currently stands. Therefore, we invite you to submit a revised version of the manuscript that addresses the points raised during the review process.

We look forward to receiving your revised manuscript.

Kind regards,

Wenjia Zhang

Academic Editor

PLOS ONE

Journal Requirements:

2. Thank you for stating the following in the Competing Interests/Financial Disclosure section* (delete as necessary):

"AP received funding from the World Bank Group via Research Agreement No. 7186069.

Funder website: https://www.worldbank.org

GJ from the sponsoring organization played a role in the study design, data collection and analysis, decision to publish and prepare the manuscript and are listed as a co-author."

We note that one or more of the authors have an affiliation to the commercial funders of this research study : 'World Bank Group'.

Please include both an updated Funding Statement and Competing Interests Statement in your cover letter. We will change the online submission form on your behalf

Additional Editor Comments (if provided):

Please carefully address the reviewers' comments. Also, the literature about methodology is quite old; and more discussions and reflections on the sequence analytic methods are needed in the final discussion section, particularly the matching and visualization techniques. Related to it, some references are suggested: Cornwell (2015) Social sequence analysis: Methods and applications. Cambridge University Press. Zhang and Thill (2017) Detecting and visualizing cohesive activity-travel patterns. Computers, Environment and Urban Systems.

Reviewers' comments:

Reviewer's Responses to Questions

**Comments to the Author**

1. Is the manuscript technically sound, and do the data support the conclusions?

Reviewer #1: Yes

Reviewer #2: Yes

2. Has the statistical analysis been performed appropriately and rigorously? 

Reviewer #1: Yes

Reviewer #2: Yes

3. Have the authors made all data underlying the findings in their manuscript fully available?

Reviewer #1: Yes

Reviewer #2: Yes

4. Is the manuscript presented in an intelligible fashion and written in standard English?

Reviewer #1: Yes

Reviewer #2: Yes

5. Review Comments to the Author

Reviewer #1: This paper aims to describe how two dimensions of Global South living standards – access to water and sanitation – have varied across the life courses of people with different types of residential history (in particular their migration experience and whether or not they have lived in a shanty town). The data are unusually rich for a study of residential moves in the Global South and the sample size is also larger than usual (n=2397). These are appealing features of the study and the findings contain some points of interest about the extent to which people can improve their residential conditions in Dar es Salaam.

However, several aspects of the manuscript require some attention before this work can be published.

1. Intro and literature review

The introduction needs to clearly explain why a longitudinal life course approach is necessary to better understand residential moves and their outcomes in Dar es Salaam. Why is this approach needed – what theoretical or practical insights are provided by looking at the entire housing history of individuals? How does this overcome the limitations of cross-sectional analyses that are (rightly in my view) critiqued in this manuscript? ‘We strongly believe…’ (p3) is an assertion that needs to be backed up by some sort of reasoned argument as to the advantages of such an approach.

Secondly, the literature review needs to more effectively explain precisely what previous studies have found about how various variables affect mobility experiences and housing career trajectories. For example, ‘push’ and ‘pull’ factors are mentioned but exactly what types of pushes and pulls are known to matter is never explained. When reviewing existing literature it is important to clearly describe what previous studies have found (e.g. by mentioning the direction of estimated effects, their magnitude, where previous research has been conducted etc). A thorough sweep of this section is required to deal with this issue.

Last, the description of life course theories is quite simplistic and contains errors – for example not all events can be considered ‘turning points’ (p6). Key aspects of the life course approach are also missing from the description such as the way this framework allows scholars to take the growing diversity, de-standardisation and dynamism of lives into account in a rich way. See Stone et al (2014) in Demography or Coulter et al (2016) in Progress in Human Geography. Furthermore, I’d like to see some discussion of period and cohort effects (see Glen Elder’s work) given that these influence the types of housing people access over the course of their lives. E.g. over time general improvements in the housing stock could be expected to increase the proportion of people living in good housing conditions.

2. Context

I think a little more contextual description is needed for a study on housing conditions. Some general background on Dar es Salaam is given but there is no description of: (i) how housing provision is organised and regulated in Tanzania; (ii) urban/development policies related to housing and water services (e.g. site and service schemes, funding mechanisms, NGO activities) or (iii) how these have changed over the times the sampled individuals have lived through. This description is really important as the opportunity structure people have experienced is probably the single most important influence on these aspects of their living conditions. E.g. if only 5% of dwellings have piped water then it is by definition going to be very unlikely that sampled households will have improved this aspect of their housing conditions over the life course. Indeed a more important question not addressed in this paper is probably who improves their housing conditions over time and who does not?

In essence, I am asking the authors to devote more space to discussing how the urban context may have enabled and/or limited how easy it is for people to improve their residential conditions over time.

3. Methods

This is a fairly basic study that describes patterns and eschews any form of explanation. This is fair enough but a major problem is that the analysis takes no account of the number of moves people have made. This means that qualitatively similar types of sequence such as 1,1,1 and 1,1,1,1 are split up and treated as separate experiences (e.g. see pp16- 17). A more data-led clustering technique or using a priori types (e.g. upward trajectory; stable; downward trajectory; downward event with recovery) would be better.

In addition, no account has been taken of the way that the age of sample members probably influences the number of moves they have made. At the very least I’d like to see some discussion of this and ideally some type of multivariate analysis should be conducted to disentangle how multiple independent variables influence housing biographies. That would yeild a richer paper.

Minor points

P3 – I’m not convinced that Global North research does tend to see residential mobility as a good thing. There are large bodies of work on residential insecurity and the way life events adversely trigger residential relocation. Take care.

P6 and elsewhere – the use of the term ‘households’ is problematic throughout this paper. A key life course insight is that households are not unified social actors that are stable through time. The composition of households changes across time and the discussion needs to reflect this. Individuals are a more appropriate unit of analysis for mobility work.

P7-8 – these hypotheses are not testable predictions and so need to be reformulated or framed as ‘expectations’.

Figures – these lack informative legends and are quite messy and unhelpful. Can the presentation be improved to make the central message easier to follow?

Reviewer #2: This is an interesting and well-written paper investigating the impact of residential mobility on housing conditions. The paper applies a novel methodology, the life course approach, and presents empirical findings from a large-scale survey recently carried out in Dar es Salaam.

My main critique of the paper is the narrow conceptualization of housing quality. The paper focus exclusively on two particular services; water and sanitation. The paper largely overlooks other important aspects of housing quality, such as space, privacy, building materials, security measures etc. The narrow focus on water and sanitation risk overlooking that residential mobility may be associated with improvements in other aspects of housing quality. The paper also overlooks wider neighborhood characteristics such as density, greenery, plot layout, security, social profile of neighbors etc. that might drive residential mobility. For an example, a household might move from a small room in a centrally located and relatively well-serviced neighborhood to a larger accommodation in a peripheral and poorly serviced neighborhood. Such a move may result in poorer access to water and sanitation, alongside improvements in relation to living space, privacy and a more pleasant and secure neighborhood. The narrow focus on water and sanitation is especially problematic in the context of Dar es Salaam, where access to water and sanitation is very limited for the city as a whole and especially beyond the central parts of the city. Therefore, access to water and sanitation may not be strongly correlated with alternative indicators of housing quality.

The narrow focus on water and sanitation also overlooks the significance of tenure form and tenure transitions as a motivation for residential mobility. For an example a new migrants in the city might start out living with relatives in the city before moving into their own rented room or house. Such a move could easily entail downgrading in relation to access to services, but might improve privacy and feeling of independence. The transition from renting to owner-occupier might also drive much residential mobility within the city. For most aspiring homeowners in Dar es Salaam this transition involves moving out of rental accommodation in consolidated central areas with relatively good access to services to engage in incremental construction of owner-occupier housing in newly developing, peripheral areas, which are often poorly serviced. As such, aspiring homeowners may have to accept poorer access to services to enjoy the benefits of owning their own home and improvements in access to services are unrelated to residential mobility, but rather linked with in-situ improvements through gradual private investments and lobbying efforts towards urban authorities. See Andreasen, MH & Agergaard, J 2016, 'Residential Mobility and Homeownership in Dar es Salaam', Population and Development Review, bind 42, no. 1, p. 95 110. https://doi.org/10.1111/j.1728-4457.2016.00104.x and Andreasen, MH & Møller-Jensen, L 2016, 'Beyond the networks: Self-help services and post-settlement network extensions in the periphery of Dar es Salaam', Habitat International, vol 53, p. 39-47. https://doi.org/10.1016/j.habitatint.2015.11.003.

Finally, the paper is characterized by unclear conceptualization of other central concepts applied in the analysis:

• Classes of households: The paper hypothesizes that living conditions vary between “classes of households” (ll. 170-171). It is unclear how classes are conceptualized and differences between classes seem not to be addressed in the results section at all.

• Migrants and non-migrants: The paper hypothesizes that the experiences of migrants are systematically different from that of non-migrants and natives (ll. 171-172). It is unclear how migrants and non-migrants are defined in the analysis. Migrants likely originate from outside the city, but how long must a person have lived in the city to no longer be conceived of as a migrant?

• Slums and non-slums: The paper hypothesizes that people living in slums follow different housing careers than those living in non-slums (ll. 172-173). It is unclear how slums (or shanties as it is called later in the paper) are defined. The paper briefly explains that slums and non-slums are identified using very high resolution satellite images (l. 210), but is it not clear on what observable characteristics are used to distinguish. The slum concept usually denotes urban areas characterized by poor housing quality, but surely this is not easily observable through analysis of satellite imagery.

Small things

• l. 185: The sweeping generalization hat Dar es Salaam has the same “environmental and infrastructural challenges similar to any other city in the world” is puzzling and would seem to require a more thorough explanation and argumentation.

6. PLOS authors have the option to publish the peer review history of their article (what does this mean?). If published, this will include your full peer review and any attached files.

Reviewer #1: No

Reviewer #2: Yes: Manja Hoppe Andreasen

---

## [Author Response · Author response to Decision Letter 0]

20 Jul 2020

Response to Reviewers:

Academic Editor’s comments:

Comment 1: Additional Editor Comments (if provided):

Please carefully address the reviewers' comments. 

Response: We are thankful to both reviewers’ very constructive feedback, and we have incorporated changes in our revised manuscript thoroughly, as outlined in this response to reviewers’ document. 

Comment 2: Also, the literature about methodology is quite old; and more discussions and reflections on the sequence analytic methods are needed in the final discussion section, particularly the matching and visualization techniques. Related to it, some references are suggested: Cornwell (2015) Social sequence analysis: Methods and applications. Cambridge University Press. Zhang and Thill (2017) Detecting and visualizing cohesive activity-travel patterns. Computers, Environment and Urban Systems.

Response: We are thankful for your very helpful suggestion. We have benefitted immensely from the resources that you have shared. We have a much better understanding of sequence analysis methods and recent developments in the network approach to social sequences now. We believe this is reflected in our revised manuscript, particularly in the Methods and the Discussions sections.

Reviewers’ Responses to Questions 

1. Is the manuscript technically sound, and do the data support the conclusions?

Reviewer #1: Yes

Reviewer #2: Yes

Authors’ Response: Thank you. 

2. Has the statistical analysis been performed appropriately and rigorously? 

Reviewer #1: Yes

Reviewer #2: Yes

Authors’ Response: Thank you.

3. Have the authors made all data underlying the findings in their manuscript fully available?

Reviewer #1: Yes

Reviewer #2: Yes

Authors’ Response: Thank you.

4. Is the manuscript presented in an intelligible fashion and written in standard English?

Reviewer #1: Yes

Reviewer #2: Yes

 Authors’ Response: Thank you.

Response to Reviewers’ Detailed Comments:

Comments from Reviewer 1 

Comment 1: This paper aims to describe how two dimensions of Global South living standards – access to water and sanitation – have varied across the life courses of people with different types of residential history (in particular their migration experience and whether or not they have lived in a shanty town). The data are unusually rich for a study of residential moves in the Global South and the sample size is also larger than usual (n=2397). These are appealing features of the study and the findings contain some points of interest about the extent to which people can improve their residential conditions in Dar es Salaam.

Response: Thank you very much for your kind words and very constructive feedback on our manuscript. 

Comment 2: However, several aspects of the manuscript require some attention before this work can be published.

1. Intro and literature review

The introduction needs to clearly explain why a longitudinal life course approach is necessary to better understand residential moves and their outcomes in Dar es Salaam. Why is this approach needed – what theoretical or practical insights are provided by looking at the entire housing history of individuals? How does this overcome the limitations of cross-sectional analyses that are (rightly in my view) critiqued in this manuscript? ‘We strongly believe…’ (p3) is an assertion that needs to be backed up by some sort of reasoned argument as to the advantages of such an approach.

Response: Thank you very much for this wonderful suggestion. We have revised our Introduction section to clearly explain why a longitudinal life-course approach is necessary and provided reasoning in a paragraph that highlights the advantages of such an approach. 

Comment 3: Secondly, the literature review needs to more effectively explain precisely what previous studies have found about how various variables affect mobility experiences and housing career trajectories. For example, ‘push’ and ‘pull’ factors are mentioned but exactly what types of pushes and pulls are known to matter is never explained. When reviewing existing literature, it is important to clearly describe what previous studies have found (e.g. by mentioning the direction of estimated effects, their magnitude, where previous research has been conducted etc). A thorough sweep of this section is required to deal with this issue.

Response: We are very grateful for this constructive suggestion. We have rewritten the Literature Review section to make it more precise and pertinent to our study. Main point we make is that such studies are non-existent in developing countries context, a gap that our exploratory study plans to fill.

Comment 4: Last, the description of life course theories is quite simplistic and contains errors – for example not all events can be considered ‘turning points’ (p6). Key aspects of the life course approach are also missing from the description such as the way this framework allows scholars to take the growing diversity, de-standardisation and dynamism of lives into account in a rich way. See Stone et al (2014) in Demography or Coulter et al (2016) in Progress in Human Geography. Furthermore, I’d like to see some discussion of period and cohort effects (see Glen Elder’s work) given that these influence the types of housing people access over the course of their lives. E.g. over time general improvements in the housing stock could be expected to increase the proportion of people living in good housing conditions.

Response: Thank you very much for this wonderful suggestion and sharing the resources on this subject. We have rewritten the section on life-course theories to cover key aspects and have made the discussion richer by incorporating the resources that you have shared. 

Comment 5: 2. Context

I think a little more contextual description is needed for a study on housing conditions. Some general background on Dar es Salaam is given but there is no description of: (i) how housing provision is organized and regulated in Tanzania; (ii) urban/development policies related to housing and water services (e.g. site and service schemes, funding mechanisms, NGO activities) or (iii) how these have changed over the times the sampled individuals have lived through. This description is really important as the opportunity structure people have experienced is probably the single most important influence on these aspects of their living conditions. E.g. if only 5% of dwellings have piped water then it is by definition going to be very unlikely that sampled households will have improved this aspect of their housing conditions over the life course. Indeed a more important question not addressed in this paper is probably who improves their housing conditions over time and who does not?

In essence, I am asking the authors to devote more space to discussing how the urban context may have enabled and/or limited how easy it is for people to improve their residential conditions over time.

Response: Thank you very much for providing this wonderful suggestion. We have rewritten the section on the case study context to provide background on the opportunity structures on the housing aspects that we study, based on an extensive review of the published literature. This has also helped us to interpret our results in the light of this context, which is a major improvement in our analysis and interpretation. We are really grateful to you for this suggestion.

Comment 6: 3. Methods

This is a fairly basic study that describes patterns and eschews any form of explanation. This is fair enough but a major problem is that the analysis takes no account of the number of moves people have made. This means that qualitatively similar types of sequence such as 1,1,1 and 1,1,1,1 are split up and treated as separate experiences (e.g. see pp16- 17). A more data-led clustering technique or using a priori types (e.g. upward trajectory; stable; downward trajectory; downward event with recovery) would be better.

In addition, no account has been taken of the way that the age of sample members probably influences the number of moves they have made. At the very least I’d like to see some discussion of this and ideally some type of multivariate analysis should be conducted to disentangle how multiple independent variables influence housing biographies. That would yeild a richer paper.

Response: Thank you very much for this very important suggestion. We have revised our analyses substantially in order to treat time in a more explicit way than the basic analyses we presented in our previous version. In particular, we have added how improvements in each of the four housing dimensions have occurred according to a decadal timeframe, added a multimedia visualization of residential mobility patterns over time using Gapminder tools, created a normalized score that captures all four dimensions in a single housing quality index, and included the number of years that each individual may have lived in a particular housing condition on all four dimensions (we have added two new dimensions following a very helpful suggestion by Reviewer 2). We believe that the new analyses and visualizations have made our descriptions much richer. We have also used a priori categories (thank you for suggesting many such categories as upward mobility, downward mobility etc. that we now use) to describe groupings that are analytically meaningful in our sequence patterns. 

Comment 7: Minor points

P3 – I’m not convinced that Global North research does tend to see residential mobility as a good thing. There are large bodies of work on residential insecurity and the way life events adversely trigger residential relocation. Take care.

Response: We have made a change on page 3 as suggested and have also thoroughly reviewed our manuscript to identify such generalizations and carefully reworded them to present a balanced view. 

Comment 8: P6 and elsewhere – the use of the term ‘households’ is problematic throughout this paper. A key life course insight is that households are not unified social actors that are stable through time. The composition of households changes across time and the discussion needs to reflect this. Individuals are a more appropriate unit of analysis for mobility work.

Response: Thank you very much for this important point. We completely agree with this view and have changed the descriptions in the entire manuscript to reflect that these trajectories are of the current household heads who may have seen changes in their own household compositions over the course of their reported histories. We were earlier driven by the literature in housing studies, where housing is commonly viewed as a household’s condition (especially when studies are cross-sectional, and hence an individual and her household have identical housing conditions). We completely agree with the important point you have made about the key life-course insight and have referred to individuals thoroughly in our revision. 

Comment 9: P7-8 – these hypotheses are not testable predictions and so need to be reformulated or framed as ‘expectations’.

Response: Thank you very much for pointing this out. We have removed the word “hypotheses” from the title of this section and elsewhere within the section to avoid suggesting testable statistical hypotheses.

Comment 10: Figures – these lack informative legends and are quite messy and unhelpful. Can the presentation be improved to make the central message easier to follow?

Response: Thank you very much for pointing this out. We have revised our figures and added a description in the main text on what they convey. Both sequence index plots and parallel coordinate plots are not commonly used outside sequence analysis literature and hence we agree that readers will benefit from such description. Thank you for pointing this out. 

Comments from Reviewer 2

Comment 1: This is an interesting and well-written paper investigating the impact of residential mobility on housing conditions. The paper applies a novel methodology, the life course approach, and presents empirical findings from a large-scale survey recently carried out in Dar es Salaam.

Response: Thank you very much for your kind words and very constructive feedback on our manuscript.

Comment 2: My main critique of the paper is the narrow conceptualization of housing quality. The paper focus exclusively on two particular services; water and sanitation. The paper largely overlooks other important aspects of housing quality, such as space, privacy, building materials, security measures etc. The narrow focus on water and sanitation risk overlooking that residential mobility may be associated with improvements in other aspects of housing quality. 

Response: Thank you very much for pointing out this limitation and suggesting other aspects of housing conditions. We have now added durable structure, which analyzes building materials, and form of tenure to our analyses. Some of the variables that you have suggested are very interesting and important, but unfortunately the survey did not collect data on all aspects of housing quality, such as space and privacy, for the entire residential history. We have added these points as future directions in our discussion section.

Comment 3: The paper also overlooks wider neighborhood characteristics such as density, greenery, plot layout, security, social profile of neighbors etc. that might drive residential mobility. For an example, a household might move from a small room in a centrally located and relatively well-serviced neighborhood to a larger accommodation in a peripheral and poorly serviced neighborhood. Such a move may result in poorer access to water and sanitation, alongside improvements in relation to living space, privacy and a more pleasant and secure neighborhood. 

Response: Thank you very much for suggesting this very important limitation in our manuscript. We have now conducted new analyses that clearly identify trade-offs between various housing conditions with each residential move. Unfortunately, we do not have data on all aspects of housing related to neighborhood characteristics for the entire history, but the new analyses capture the four housing dimensions for which data was available. We have also suggested collecting data on neighborhood characteristics in future studies as part of our discussion section. 

Comment 4: The narrow focus on water and sanitation is especially problematic in the context of Dar es Salaam, where access to water and sanitation is very limited for the city as a whole and especially beyond the central parts of the city. Therefore, access to water and sanitation may not be strongly correlated with alternative indicators of housing quality.

Response: Thank you for this suggestion. As noted earlier, we have added two more dimensions of housing quality, namely type of building materials (durable structure) and ownership form (tenure security). In addition, we have created a housing quality index that combines all four dimensions and normalizes it across time to capture the entire lived experience in a single score for each and every individual (up to 2014, when data was collected). Thanks to this suggestion, a holistic approach to housing quality (limited only by data availability) has strengthened our analyses.

Comment 5: The narrow focus on water and sanitation also overlooks the significance of tenure form and tenure transitions as a motivation for residential mobility. For an example a new migrants in the city might start out living with relatives in the city before moving into their own rented room or house. Such a move could easily entail downgrading in relation to access to services, but might improve privacy and feeling of independence. The transition from renting to owner-occupier might also drive much residential mobility within the city. For most aspiring homeowners in Dar es Salaam this transition involves moving out of rental accommodation in consolidated central areas with relatively good access to services to engage in incremental construction of owner-occupier housing in newly developing, peripheral areas, which are often poorly serviced. As such, aspiring homeowners may have to accept poorer access to services to enjoy the benefits of owning their own home and improvements in access to services are unrelated to residential mobility, but rather linked with in-situ improvements through gradual private investments and lobbying efforts towards urban authorities. See Andreasen, MH & Agergaard, J 2016, 'Residential Mobility and Homeownership in Dar es Salaam', Population and Development Review, bind 42, no. 1, p. 95 110. https://doi.org/10.1111/j.1728-4457.2016.00104.x and Andreasen, MH & Møller-Jensen, L 2016, 'Beyond the networks: Self-help services and post-settlement network extensions in the periphery of Dar es Salaam', Habitat International, vol 53, p. 39-47. https://doi.org/10.1016/j.habitatint.2015.11.003.

Response: Thank you very much for raising this very important point and sharing very valuable resources. We now explicitly incorporate ownership in our analyses, as well as the trade-offs that are involved in residential mobility. Our multimedia files also capture such trade-offs in services with ownership in a dynamic visualization. The articles you have shared with us have provided us with very important insights into residential mobility in Dar es Salaam and have informed all of our new analyses. We are deeply grateful to you for pointing us to these resources. 

Comment 6: Finally, the paper is characterized by unclear conceptualization of other ce

ntral concepts applied in the analysis:

• Classes of households: The paper hypothesizes that living conditions vary between “classes of households” (ll. 170-171). It is unclear how classes are conceptualized and differences between classes seem not to be addressed in the results section at all.

Response: Thank you for pointing this out. We have removed the word “classes” to avoid confusion. We were simply referring to migrants and natives, which is immediately clear without using the word “classes.”

Comment 7: 

• Migrants and non-migrants: The paper hypothesizes that the experiences of migrants are systematically different from that of non-migrants and natives (ll. 171-172). It is unclear how migrants and non-migrants are defined in the analysis. Migrants likely originate from outside the city, but how long must a person have lived in the city to no longer be conceived of as a migrant?

Response: We have only considered the place of birth in defining migration status. We have added this definition in our description of migrants in the Data section. Since we examine the entire history, using cut-off time as a defining factor will create uneven histories (some houses were occupied as migrants while other houses were occupied as non-migrants as individuals assimilate in the city over time). We understand that it is a rather simplistic approach, but we believe that it still provides a meaningful comparison with those who are born in Dar es Salaam. 

Comment 8: 

• Slums and non-slums: The paper hypothesizes that people living in slums follow different housing careers than those living in non-slums (ll. 172-173). It is unclear how slums (or shanties as it is called later in the paper) are defined. The paper briefly explains that slums and non-slums are identified using very high resolution satellite images (l. 210), but is it not clear on what observable characteristics are used to distinguish. The slum concept usually denotes urban areas characterized by poor housing quality, but surely this is not easily observable through analysis of satellite imagery.

Response: We have used the indicator that has been developed with the use of high-resolution satellite images by the data collection team. The data collection used the imagery to create strata for the survey design. We now point readers to those details that are provided in the documentation, which is released along with the publicly available data. We simply use the indicator that is supplied with the data—we did not create it ourselves. We have now made this point clear in Data section. Thank you very much for pointing out this source of confusion. 

Comment 9: Small things

• l. 185: The sweeping generalization hat Dar es Salaam has the same “environmental and infrastructural challenges similar to any other city in the world” is puzzling and would seem to require a more thorough explanation and argumentation.

Response: We have removed this generalization here and thoroughly reviewed our manuscript for such generalizations and either removed them (if they are not central to the thesis of our paper, such as in this instance) or provided a thorough explanation (if they are important to the thesis of this manuscript).

---

## [Decision Letter · Decision Letter 1]

18 Aug 2020

PONE-D-20-11700R1

Effects of Residential Mobility and Migration on Standards of Living in Dar es Salaam, Tanzania: A Life-course Approach

PLOS ONE

Dear Dr. Patel,

Thank you for submitting your manuscript to PLOS ONE. After careful consideration, we feel that it has merit but does not fully meet PLOS ONE’s publication criteria as it currently stands. Therefore, we invite you to submit a revised version of the manuscript that addresses the points raised during the review process.

We look forward to receiving your revised manuscript.

Kind regards,

Wenjia Zhang

Academic Editor

PLOS ONE

Reviewers' comments:

Reviewer's Responses to Questions

**Comments to the Author**

1. If the authors have adequately addressed your comments raised in a previous round of review and you feel that this manuscript is now acceptable for publication, you may indicate that here to bypass the “Comments to the Author” section, enter your conflict of interest statement in the “Confidential to Editor” section, and submit your "Accept" recommendation.

Reviewer #1: (No Response)

Reviewer #2: All comments have been addressed

2. Is the manuscript technically sound, and do the data support the conclusions?

Reviewer #1: Yes

Reviewer #2: Yes

3. Has the statistical analysis been performed appropriately and rigorously? 

Reviewer #1: Yes

Reviewer #2: Yes

4. Have the authors made all data underlying the findings in their manuscript fully available?

Reviewer #1: Yes

Reviewer #2: Yes

5. Is the manuscript presented in an intelligible fashion and written in standard English?

Reviewer #1: Yes

Reviewer #2: Yes

6. Review Comments to the Author

Reviewer #1: This is my second reading of this manuscript and it is pleasing to see that the authors have attended to the issues raised in my previous review. A better case is built for doing a longitudinal analysis and the discussion of life course perspectives is much improved. I also appreciate the use of a priori sequence ‘types’ that give much greater clarity to the analysis. Taken together, these changes have produced a much-improved paper. Good work.

There are a few small changes I’d suggest the authors consider in their revision:

P3 - can you give a local example or two of the types of policies mentioned here?

P5 – it would be worth mentioning the gendered nature of migration here. Developed world research has identified women as ‘tied movers/stayers’ (Cooke 2008 in Population, Space and Place) and the issue of ‘left-behind’ families is a key one across much of the Global South. The basic point to note here is that the positive wage benefits of migrating may accrue only to certain household members.

P6 – Household ageing is a problematic idea when households are made up of varying persons. Try and make sure the paper consistently notes that your analysis is of individual life course and housing careers.

P9 – RQ2 implies causality and that cannot be tested here. Avoid causal language unless justified by your research approach (e.g. the use of experimental rather than observational data).

Sequence plots – I still find these rather messy and unhelpful. For example, I struggle to see from them what the most common patterns of sequence are (as described in the text). Can they be improved in some way?

Sequence plots – the n here looks to be <2000 for most of the figures. This makes sense when subgroup analyses are conducted but not when whole sample descriptives are presented (e.g. Figures 1-2). Why is the n < 2397?

Migrant and shanty sequence analysis – rather than presenting so many plots, I wonder whether crosstabs of migrant/shanty status (x) against sequence type (y) would be easier to read? They would also enable statistical comparisons to be drawn – e.g. to see whether more migrants than stayers are ‘perpetually deprived’.

Figure 14 is rather hard to read.

Reviewer #2: The authors have addressed my comments comprehensively and within the limitations of the data. The authors have undertaken significant revisions which have greatly improved the paper. Generally, I think this manuscript is ready for publication. I have a couple of small comments:

l. 221: Dar es Salaam is not home to 40% of Tanzania's population. The total population of Tanzania was 44.9 million according to the 2012 census, whereas in Dar es Salaam region it was 4.4 million

l. 264: Why the date and time stamp in the text?

7. PLOS authors have the option to publish the peer review history of their article (what does this mean?). If published, this will include your full peer review and any attached files.

Reviewer #1: **Yes: **Rory Coulter

Reviewer #2: No

---

## [Author Response · Author response to Decision Letter 1]

25 Aug 2020

Response to Reviewers

Reviewers’ Responses to Questions 

1. If the authors have adequately addressed your comments raised in a previous round of review and you feel that this manuscript is now acceptable for publication, you may indicate that here to bypass the “Comments to the Author” section, enter your conflict of interest statement in the “Confidential to Editor” section, and submit your "Accept" recommendation.

Reviewer #1: (No Response)

Reviewer #2: All comments have been addressed

Authors’ Response: Thank you. 

2. Is the manuscript technically sound, and do the data support the conclusions?

Reviewer #1: Yes

Reviewer #2: Yes

Authors’ Response: Thank you.

3. Has the statistical analysis been performed appropriately and rigorously? 

Reviewer #1: Yes

Reviewer #2: Yes

Authors’ Response: Thank you.

4. Have the authors made all data underlying the findings in their manuscript fully available?

Reviewer #1: Yes

Reviewer #2: Yes

Authors’ Response: Thank you.

5. Is the manuscript presented in an intelligible fashion and written in standard English?

Reviewer #1: Yes

Reviewer #2: Yes

7. PLOS authors have the option to publish the peer review history of their article (what does this mean?). If published, this will include your full peer review and any attached files. If you choose “no”, your identity will remain anonymous, but your review may still be made public. Do you want your identity to be public for this peer review? For information about this choice, including consent withdrawal, please see our Privacy Policy.

Reviewer #1: Yes: Rory Coulter

Reviewer #2: No

Authors’ Response: Thank you for your time and very careful review of both the versions of this manuscript. We cannot thank you enough for your constructive feedback.

Response to Reviewers’ Detailed Comments:

Comments from Reviewer 1 

Comment 1: This is my second reading of this manuscript and it is pleasing to see that the authors have attended to the issues raised in my previous review. A better case is built for doing a longitudinal analysis and the discussion of life course perspectives is much improved. I also appreciate the use of a priori sequence ‘types’ that give much greater clarity to the analysis. Taken together, these changes have produced a much-improved paper. Good work.

Response: Thank you very much for carefully reviewing our manuscript and providing very constructive feedback – twice. We really appreciate it. 

Comment 2: There are a few small changes I’d suggest the authors consider in their revision:

P3 - can you give a local example or two of the types of policies mentioned here?

Response: Thank you very much for this wonderful suggestion. We have added a new discussion to substantiate our point on temporal dimensions and provided a reference that has multiple examples of spatially oriented slum policies.

Comment 3: P5 – it would be worth mentioning the gendered nature of migration here. Developed world research has identified women as ‘tied movers/stayers’ (Cooke 2008 in Population, Space and Place) and the issue of ‘left-behind’ families is a key one across much of the Global South. The basic point to note here is that the positive wage benefits of migrating may accrue only to certain household members.

Response: We are grateful for this constructive suggestion. Thank you for sharing the resources on this subject. We have now added these points in our discussion. 

Comment 4: P6 – Household ageing is a problematic idea when households are made up of varying persons. Try and make sure the paper consistently notes that your analysis is of individual life course and housing careers.

Response: Thank you very much for pointing this out. We have thoroughly reviewed our manuscript to make sure that we consistently use individuals/people as opposed to households and household members.

Comment 5: P9 – RQ2 implies causality and that cannot be tested here. Avoid causal language unless justified by your research approach (e.g. the use of experimental rather than observational data).

Response: Thank you very much for providing this wonderful suggestion. We agree that our research approach does not warrant a discussion on causality. We have revised the second research question to ensure that it is in congruence with our analysis.

Comment 6: Sequence plots – I still find these rather messy and unhelpful. For example, I struggle to see from them what the most common patterns of sequence are (as described in the text). Can they be improved in some way?

Response: Thank you very much for your suggestion. We believe that part of the problem is that our data is not a balanced panel (with varying numbers of housing careers resulting in varying lengths of history), unlike a typical use case for sequence index plots. Second, the graphs present a rather large sample (many textbook examples we checked have fewer individuals). We have used parallel coordinate plots in conjunction with sequence index plots to aid with visualization of the most common patterns. We also added a description in the captions of these figures to clarify what each of these panels represent in our earlier revision. We agree that figures have limited utility, but tables provide the frequency of the most common patterns with numeric precision to overcome that limitation.

Comment 7: Sequence plots – the n here looks to be <2000 for most of the figures. This makes sense when subgroup analyses are conducted but not when whole sample descriptives are presented (e.g. Figures 1-2). Why is the n < 2397?

Response: Thank you for pointing this out. Sequences with gaps in history in a variable of interest (e.g., water) are excluded from the analysis. We have clearly identified this limitation in the data section and reported missing values in residential history. These gaps in history explain the lower N in sequence plots.

Comment 8: Migrant and shanty sequence analysis – rather than presenting so many plots, I wonder whether crosstabs of migrant/shanty status (x) against sequence type (y) would be easier to read? They would also enable statistical comparisons to be drawn – e.g. to see whether more migrants than stayers are ‘perpetually deprived’.

Response: Thank you very much for this excellent point. We have added the suggested tables that enable statistical comparison. We have revised the text to refer to the content of the table. We have moved Figures 5 to 12 to the supplemental materials section and referred them in our revised text.

Comment 9: Figure 14 is rather hard to read.

Response: Thank you very much for pointing this out. We have increased the font size of graphs and improved the layout. We believe that the system-generated PDF does not include high-resolution images. We have made sure that the image is readable in its final form. 

Comments from Reviewer 2

Comment 1: The authors have addressed my comments comprehensively and within the limitations of the data. The authors have undertaken significant revisions which have greatly improved the paper. Generally, I think this manuscript is ready for publication. 

Response: Thank you very much for your kind words and very constructive feedback on our manuscript.

Comment 2: I have a couple of small comments:

l. 221: Dar es Salaam is not home to 40% of Tanzania's population. The total population of Tanzania was 44.9 million according to the 2012 census, whereas in Dar es Salaam region it was 4.4 million

Response: Thank you very much for pointing out this error. We missed writing Tanzania’s urban population. We have corrected it now and cited the source of this information. 

Comment 3: l. 264: Why the date and time stamp in the text? 

Response: Thank you very much for pointing this out. Our reference management tool, Zotero, sometimes erroneously replaces cited references with a time stamp. In this version, we have removed all automatically updating fields to make sure that our final version doesn’t have any such typographical errors.

---

## [Decision Letter · Decision Letter 2]

14 Sep 2020

Effects of Residential Mobility and Migration on Standards of Living in Dar es Salaam, Tanzania: A Life-course Approach

PONE-D-20-11700R2

Dear Dr. Patel,

We’re pleased to inform you that your manuscript has been judged scientifically suitable for publication and will be formally accepted for publication once it meets all outstanding technical requirements.

Kind regards,

Wenjia Zhang

Academic Editor

PLOS ONE

Additional Editor Comments (optional):

Reviewers' comments:

Reviewer's Responses to Questions

**Comments to the Author**

1. If the authors have adequately addressed your comments raised in a previous round of review and you feel that this manuscript is now acceptable for publication, you may indicate that here to bypass the “Comments to the Author” section, enter your conflict of interest statement in the “Confidential to Editor” section, and submit your "Accept" recommendation.

Reviewer #1: All comments have been addressed

Reviewer #2: All comments have been addressed

2. Is the manuscript technically sound, and do the data support the conclusions?

Reviewer #1: (No Response)

Reviewer #2: Yes

3. Has the statistical analysis been performed appropriately and rigorously? 

Reviewer #1: (No Response)

Reviewer #2: Yes

4. Have the authors made all data underlying the findings in their manuscript fully available?

Reviewer #1: (No Response)

Reviewer #2: Yes

5. Is the manuscript presented in an intelligible fashion and written in standard English?

Reviewer #1: (No Response)

Reviewer #2: Yes

6. Review Comments to the Author

Reviewer #1: (No Response)

Reviewer #2: (No Response)

7. PLOS authors have the option to publish the peer review history of their article (what does this mean?). If published, this will include your full peer review and any attached files.

Reviewer #1: **Yes: **Rory Coulter

Reviewer #2: No

---

## [Editor Report · Acceptance letter]

18 Sep 2020

PONE-D-20-11700R2 

Effects of Residential Mobility and Migration on Standards of Living in Dar es Salaam, Tanzania: A Life-course Approach 

Dear Dr. Patel:

I'm pleased to inform you that your manuscript has been deemed suitable for publication in PLOS ONE. Congratulations! Your manuscript is now with our production department. 

Kind regards, 

on behalf of

Dr. Wenjia Zhang 

Academic Editor

PLOS ONE